# High-dimensional Continuum Armed and High-dimensional Contextual Bandit: with Applications to Assortment and Pricing

## Abstract

The bandit problem with high-dimensional continuum arms and high-dimensional contextual covariates is often faced by decision-makers but remains unsolved. Recent developments in contextual bandit problems focus on the setting where the number of arms are small but are impracticable with high-dimensional continuous arm spaces. To bridge the gap, we propose a novel model for the high-dimensional continuum armed and high-dimensional contextual bandit problem that captures the effect of the arm and covariates on the reward via a low-rank representation matrix. The representation matrix is endowed with interpretability and predictive power. We further propose an efficient bandit algorithm based on a low-rank matrix estimator with theoretical justifications. The generality of our model allows wide applications including business and healthcare. In particular, we apply our method to assortment and pricing, both of which are important decisions for firms such as online retailers. Our method can solve the assortment-pricing problem simultaneously while most existing methods address them separately. We demonstrate the effectiveness of our method to jointly optimize assortment and pricing for revenue maximization for a giant online retailer.

## 1 Introduction

The bandit problem dates back to when Robbins (1952) formulated the problem as the sequential design of experiments and has been studied to a great extent recently due to the demand for online decision-making, especially from e-commerce and health care. A decision-maker chooses an action (arm) at each round and observes a reward and the goal is to act strategically so as to find an optimal action that maximizes the long-term reward without sacrificing too much.

The bandit literature mostly focuses on the problem of a finite number of independent arms, but it is often the case that infinite number of of arms and the arms share some common structure and thus can be indexed by variables as a continuum armed bandit problem. In e-commerce, the retailer needs to decide the product assortment and pricing to maximize long-term profits; in mobile health, the personal device provides exercise and dietary suggestions to improve physical and mental health. The possible actions in both examples can be parameterized as continuous variables, which are possibly high dimensional. In addition, decision-makers observe other covariates/features, i.e., the contextual bandit problem where the reward is modeled as a function of unknown parameters and the contextual variables, and in many practical settings, the covariates are high-dimensional.

As the dimensionalities of the action space (for arms) and the contextual variables grow, the traditional bandit algorithms suffer from the curse of dimensionality and it is impossible or prohibitively costly to learn the optimal decision. Albeit both the arm and the contextual are high-dimensional, the dimension of the underlying factors is often, fortunately, small – for high-dimensional bandit problems, one can assume a low-dimensional structure on the unknown parameters, such as the LASSO bandit (Bastani & Bayati, 2020); and for high-dimensional continuum armed bandit problems, one can assume the reward function depends only on the low-dimensional subspace of the action space (Tyagi et al., 2016). While low-dimensional representation has been successfully adopted in high-dimensional bandit problems and high-dimensional continuum armed bandit problems respectively, a natural but important question remains open: *can we efficiently solve the bandit problem with both high-dimensional continuum arms and high-dimensional contextual variables simultaneously?*

In this paper, we tackle the above problem by proposing a novel model that captures the effect of the arm and the contextual with an approximately low-rank matrix representation as well as an efficient algorithm (`Hi-CCAB`) to efficiently solve the problem with theoretical justifications. Specifically, for an action that is presented as a vector $\boldsymbol{a} \in \mathbb{R}^{d_a}$ and the corresponding contextual covariates $\boldsymbol{x} \in \mathbb{R}^{d_x}$, we yield reward

$$r = \boldsymbol{a}^\top \boldsymbol{\Theta} \boldsymbol{x} + \varepsilon$$

where $\boldsymbol{\Theta} \in \mathbb{R}^{d_a \times d_x}$ is the unknown representation matrix, which is assumed to have rank $d \ll \min\{d_a, d_x\}$ and $\varepsilon$ is the independent error. To learn the low-rank representation matrix, we adapt the low-rank matrix estimator to the bandit setting. We further demonstrate the benefits of our methodologies in e-commerce with real sales data where the online retailer needs to decide on the product assortment and pricing jointly. The generality of our model makes it possible to learn policy on product assortment and pricing at the same time, while previous literature mostly studies the assortment and pricing problem separately.

**Contributions.** We highlight the following contributions of our paper:

1. We propose a new model for high-dimensional continuum armed and high-dimensional contextual bandit problem, which is often faced by decision-makers but very little existing literature attempts to solve. The crux of our model is the low-rank representation matrix that exploits the low-dimensional structure of both the high-dimensional arms and high-dimensional covariates. Our model unifies a large class of bandit models.

2. The low-rank representation matrix is endowed with interpretability and predictive power. One can perform singular value decomposition (SVD) on the representation matrix – the left singular vectors reveal the latent structure and relationships among the arms, while the right singular vectors show the latent factors of the covariate. In other words, our model implicitly performs principle component analysis (PCA) on the effect of arms and covariates on the mean reward. On the other hand, given the covariate, our model is able to predict the reward of an unseen arm. Both interpretability and predictive power can be tremendously useful for decision-makers.

3. We propose an efficient algorithm for the **Hi**gh-dimensional **C**ontextual and **Hi**gh-dimensional **C**ontinumm **A**rmed **B**andit (`Hi-CCAB`) by adopting the low-rank matrix estimator. We further provide an upper bound for the convergence rate of `Hi-CCAB` in terms of the time-averaged expected cumulative regret.

4. The generality of our model allows for a wide range of applications. Specifically, we apply `Hi-CCAB` to the joint assortment and pricing problem. We show that our model reveals insights for product designs, assortment, and pricing and that the assortment-pricing policy based on `Hi-CCAB` yields sales four times as high as the original strategy.

**Literature review.** Literature on high-dimensional bandit problems has been expanding recently, especially after statistical tools for high-dimensional problems become mature (Negahban & Wainwright, 2011; Wainwright, 2019). Lots of high-dimensional bandit literature focuses on contextual bandits with high-dimensional covariates, such as the LASSO bandit problem (Abbasi-Yadkori et al., 2012; Kim & Paik, 2019; Bastani & Bayati, 2020; Hao et al., 2020; Papini et al., 2021) where they assume the mean reward is a linear function of a sparse unknown parameter vector, the low-rank matrix bandit where the covariate and unknown parameter are both of matrix form (Kveton et al., 2017; Lu et al., 2021), and other non-parametric methods that learns the reward function using random forest or deep learning (Féraud et al., 2016; Zhou et al., 2020; Ban et al., 2022; Chen et al., 2022; Xu et al., 2022). The high-dimensional bandit models are special cases of our model.

Another stream of high-dimensional bandit literature studies representation learning in linear bandits, specifically for multi-task learning where several bandits are played concurrently. The arms for each task are embedded in the same space and share a common low-dimensional representation (Lale et al., 2019; Yang et al., 2020; Hu et al., 2021; Xu & Bastani, 2021). Our problem is different from multi-task learning since at each time we only have one bandit and thus observe one reward while in the multi-task bandit problem, multiple bandits are played at the same time.

For continuum armed bandits, there exists a thread of literature that assumes the mean reward function is smooth and continuous on the action space in some sense, e.g., the function lies in the Lipschitz or Hölder space (Agrawal, 1995; Kleinberg, 2004; Kleinberg et al., 2019). Most work discretizes the arm space or adopts the non-parametric regression to estimate the reward function, which is very different from our approach. Recent literature studies on continuum armed bandit with

contextual covariates further assumes the mean reward function is continuous on the arm-covariate space (Lu et al., 2010; Slivkins, 2011; Krishnamurthy et al., 2020). Literature on high-dimensional continuum armed bandits, however, is scarce (Turğay et al., 2020; Majzoubi et al., 2020). Again, the techniques and assumptions therein are different from ours and their model is hard to interpret.

In terms of matrix estimation techniques, low-rank matrix estimation and recovery have been studied extensively in statistics and widely used in numerous applications (Candes & Plan, 2010; Candès & Tao, 2010; Negahban & Wainwright, 2011; Shabalin & Nobel, 2013; Gavish & Donoho, 2014; Cai & Zhang, 2018; Wainwright, 2019). We adapt the techniques in this literature to provide convergence analysis for our algorithm.

Finally, in operation research, assortment and pricing are important decisions for firms and there exists voluminous literature on dynamic assortment and dynamic pricing. Most of the work on assortment is based on the multinomial logit (MNL) choice model (Caro & Gallien, 2007; Kök et al., 2008; Sauré & Zeevi, 2013) and recently a strand of work adopt the multi-arm bandit technique to the MNL model (Chen & Wang, 2017; Agrawal et al., 2019; Kallus & Udell, 2020; Chen et al., 2021). For dynamic pricing, the problem usually comes with demand learning. In presence of covariates, the demand can be modeled as a parametric function (Qiang & Bayati, 2016; Ban & Keskin, 2021) or a nonparametric function (Chen & Gallego, 2021) which adopt the continuum armed bandit techniques in Slivkins (2011). However, there are relatively few papers on the joint assortment-pricing problem. Recently, Miao & Chao (2021) provides a solution using the MNL choice model with finite arms, while our model targets at infinite many arms. In addition, their model assumes the products are independent of each other and can only handle a small number of products. Their model can neither incorporates contextual information nor predicts new products.

**Roadmap.** The rest of the paper is organized as follows. Section 2 describes the problem formulation and introduces our model with two concrete examples in assortment-pricing and health care. Section 3 presents our `Hi-CCAB` algorithm and its convergence result. Finally, Section 4 shows the empirical results on simulated data and a case study on real sales data from one of the largest online retailers. The proof of our theorem and additional empirical results are provided in the Appendix.

## 2 PROBLEM FORMULATION

In this section, we first introduce our high-dimensional continuum armed and high-dimensional contextual bandit model. Since our model is novel and different from traditional bandit models, we further provide intuition and two real applications of our model in assortment-pricing and healthcare. Finally, we show that a large class of bandit models can be reformulated into our model.

**Notation.** We use bold lowercase for vectors and bold uppercase for matrices. For any vector $\boldsymbol{a}$, we use $\|\boldsymbol{a}\|$ to denote its $\ell_2$ norm. For any matrix $\boldsymbol{A}$, we use $\|\boldsymbol{A}\|_F := \sum_{ij} a_{ij}^2$ to denote its Frobenius norm, $\|\boldsymbol{A}\|_2$ to denote its $\ell_2$ spectrum norm, i.e., $\|\boldsymbol{A}\|_2 := \sup_{\|\boldsymbol{x}\|_2=1} \|\boldsymbol{A}\boldsymbol{x}\|_2$, and $\|\boldsymbol{A}\|_* := \sum_{k=1}^{d} s_k$ to denote its nuclear norm where $d$ is the rank and $s_k$'s are the singular values of $\boldsymbol{A}$. We use $\langle \boldsymbol{a}, \boldsymbol{b} \rangle := \boldsymbol{a}^\top \boldsymbol{b}$ to denote the inner product between two vectors and $\langle \boldsymbol{A}, \boldsymbol{B} \rangle :=$ trace($\boldsymbol{A}^\top \boldsymbol{B}$) between two matrices.

**Problem setup.** At each time $t$, we make one decision for a batch of objects of size $L$. Before making the decision, we observe the attributes of these $L$ objects, which can be characterized in potentially high-dimensional contextual vectors: $\boldsymbol{x}_{t,1}, \cdots, \boldsymbol{x}_{t,L} \in \mathbb{R}^{d_x}$. Then based on all the observations we have before time $t$ and the contextual vectors at time $t$, we decide on an action to take (or equivalently an arm to choose), which can be characterized as a high-dimensional vector $\boldsymbol{a}_t$ that takes value in a constraint set $\mathcal{A}$ in high-dimensional space $\mathbb{R}^{d_a}$. After we take the action each time, we observe a batch of rewards,

$$r_{t,j} = \boldsymbol{a}_t^\top \boldsymbol{\Theta} \boldsymbol{x}_{t,j} + \varepsilon_{t,j}, \quad j = 1, 2, \cdots, L \tag{1}$$

where $\boldsymbol{\Theta}$ is a *low-rank* matrix and $\varepsilon_{t,j}$ is independent noise with $\mathbb{E}[\varepsilon_{t,j}] = 0$ and $\text{Var}[\varepsilon_{t,j}] \leq \sigma^2$.

As a bandit problem, our goal is to design a sequential decision-making policy $\pi$ that maximizes the expected cumulative reward, or equivalently, minimizes the expected cumulative regret. Specifically, suppose policy $\pi$ governs the way we take actions $\boldsymbol{a}_1, \boldsymbol{a}_2, \boldsymbol{a}_3, \cdots$, we have an *expected cumulative regret* measuring the difference between the cumulative expected reward of the best possible action when the underlying true parameter (i.e., $\boldsymbol{\Theta}$) is known and that we can achieve under policy $\pi$,

$$R_T^\pi = \mathbb{E}\left[ \sum_{t=1}^{T} \max_{\boldsymbol{a} \in \mathcal{A}_t} \left( \sum_{j=1}^{L} \boldsymbol{a}^\top \boldsymbol{\Theta} \boldsymbol{x}_{t,j} - \boldsymbol{a}_{t,\pi}^\top \boldsymbol{\Theta} \boldsymbol{x}_{t,j} \right) \right] \tag{2}$$

where the expectation is taken with respect to $(\boldsymbol{x}_{t,j}, \varepsilon_{t,j})$ since $\boldsymbol{a}_{t,\pi}$ depends on both. We seek an optimal policy $\pi^*$ that minimizes the expected cumulative regret $R_T^\pi$. Note that $R_T^\pi$ grows with $T$. To better measure the performance of a policy, we focus on the *time-averaged* expected cumulative regret, $R_T^\pi/T$. We will ignore the subscript $\pi$ for $\boldsymbol{a}$ for notation simplicity in the rest of the paper.

At first sight, our model seems remote from other bandit models and hard to interpret. Our model is, in fact, a generalization of a large class of bandit models, and the generality of our model makes it applicable to a wide range of decision-making problems. In the following, we will parse the model and provide more intuitions.

Let us consider the classical $K$-arm bandit model (without context). Each arm can be represented by a $k$-dimensional standard basis and the covariate vector is simply 1. Then $\boldsymbol{\Theta}$ becomes a vector where each element is the mean reward for the corresponding arm. For the multi-arm contextual linear bandit problem, we further observe covariate $\boldsymbol{x}$ as the contextual. Then each row of $\boldsymbol{\Theta}$ becomes the coefficient $\boldsymbol{\beta}$ for each arm, i.e., $r_k = \boldsymbol{\beta}_k^\top \boldsymbol{x}$ for $k = 1, \ldots, K$. Our model further unifies a large class of bandit models and we will formalize the above statements later in Proposition 1.

The novelty of our model lies in the low-rank representation matrix $\boldsymbol{\Theta}$. It encapsulates the effect of the arm and covariates on the reward and exploits the low-dimensional structure in the high-dimensional arm and covariates. To be more specific, let us consider $\boldsymbol{\Theta}$ to be exactly low-rank and of rank $d$. Suppose its singular value decomposition is $\boldsymbol{\Theta} = \boldsymbol{USV}$, where $\boldsymbol{U}^\top \boldsymbol{U} = \boldsymbol{I}_d$, $\boldsymbol{S}$ is a $d \times d$ diagonal matrix with positive diagonal elements, and $\boldsymbol{V}^\top \boldsymbol{V} = \boldsymbol{I}_d$. Let the left singular vectors $\boldsymbol{U} = (\boldsymbol{u}_1, \ldots, \boldsymbol{u}_d)$, the singular values in the diagonal of $\boldsymbol{S}$ be $s_1 \geq s_2 \geq \ldots \geq s_d > 0$ and the right singular vectors $\boldsymbol{V} = (\boldsymbol{v}_1, \ldots, \boldsymbol{v}_d)$. Then the mean reward in (1) can be re-expressed as

$$\mathbb{E}[r] = \boldsymbol{a}^\top \boldsymbol{\Theta} \boldsymbol{x} = \sum_{i=1}^d s_i \langle \boldsymbol{a}, \boldsymbol{u}_i \rangle \cdot \langle \boldsymbol{v}_i, \boldsymbol{x} \rangle. \tag{3}$$

In other words, the mean reward is the summation of inner products between the action projected on the left singular vector and the covariates projected on the right singular vector, weighted by the singular values. By assuming $\boldsymbol{\Theta}$ to be low-rank, the mean reward is assumed to be governed by only a few linear combinations of the arm attributes and covariates. Hence our model automatically explores the low-dimensional structure of the arm vector and the contextual vector in terms of its effect on the reward, from which we can draw interpretation and insights from the effective subspaces of both the arm and covariates.

As a concrete illustration of our model and to explain why our model is reasonable in real applications, we provide the following use cases in the joint assortment-pricing problem and health care.

**Example 1** (Assortment and Pricing). *In retailing and e-commerce, the assortment problem is to decide what combination of products to present at each given time with constraints on the capacity (Kök et al., 2008), and the pricing problem is to decide the prices of the products. The goal of the two problems is to maximize certain objective such as maximizing the revenue or profit.*

*Products can be usually characterized by attributes such as color, pattern, and fit for apparels or technical specifications for electronics and appliance. We focus on instant noodles, which will be our case study in Section 4. Each product is single-flavor or assorted with different packs and can be represented as a feature vector $\tilde{\boldsymbol{p}} = (\#flavor_1, \#flavor_2, \cdots, \#flavor_m)$ where $m$ is the number of possible flavors and is priced as $p$. Then the store needs to decide on what products to present and their corresponding prices. Namely, the arm (action) vector can be represented as $\boldsymbol{a} = (\tilde{\boldsymbol{p}}_1, p_1, \tilde{\boldsymbol{p}}_2, p_2, \cdots, \tilde{\boldsymbol{p}}_K, p_K, 1)$ where $K$ is the maximum number of slots. The arm vector is clearly in a high-dimensional continuous space. At the same time, we observe the contextual covariates $\boldsymbol{x}$ for each period of time, such as the location and season at the aggregated level or demographics information at the user level.*

*The demand and sales of products with similar attributes react similarly to the same market conditions. It is often the case that there exists latent factors of the products that governs the demand and sales. Therefore, it is reasonable to parameterize the reward function in the form of (1) rather than ignoring the similarity between products as in the literature (Miao & Chao, 2021; Kallus & Udell, 2020; Chen et al., 2021). Our model can further suggests new products rather than only the products that has already been provided.*

**Example 2** (Healthcare). *In healthcare, for the health-monitoring apps which both monitors health conditions and give suggestions on actions to take for users, the arm ($\boldsymbol{a}$) is high-dimensional and continuous (e.g., sleeping time, length and kind of exercise, usage of social media, diet choices*

*including energy, water, protein, minerals, and nutrition intakes), and the health outcome not only depends on our suggestions, but also depends on the user's characteristics (e.g., age, gender, weight, height, basic health status, tendency of following suggestions) as contextual variables ($\boldsymbol{x}$). Clearly, both the arm and the contextual variable vector are possibly high-dimensional and the arm can take continuous values. The classical bandit models do not fit the situation. The actions usually share similar effect on health and the user's characteristics can be usually captured by a few latent factors. Therefore, it is reasonable to assume $\boldsymbol{\Theta}$ to be low-rank.*

To close the section, Proposition 1 shows that the traditional multi-arm bandit, multi-arm high-dimensional contextual bandit, and continuum arm bandit can be written in the form of model (1).

**Proposition 1.** *The following bandit models can be expressed as special cases of our model.*

1. *(multi-arm bandit) For $i$-th arm, $\boldsymbol{a} = (0, 0, \cdots, 1, \cdots, 0)$, where $1$ is in $i$-th element. Suppose $\boldsymbol{x}$ has its first element being constant. Then $\boldsymbol{\Theta}_{i,1} = \mu_i$, where $\mu_i$ is the mean reward of the $i$-th arm, and $\boldsymbol{\Theta}_{i,j} = 0$ if $j \neq 1$. Clearly, $\boldsymbol{\Theta}$ has rank 1.*

2. *(multi-arm high-dimensional contextual bandit) For $i$-th arm, $\boldsymbol{a} = (0, 0, \cdots, 1, \cdots, 0)$, where $1$ is in $i$-th element. $\boldsymbol{x}$ is the contextual vector. Then $\boldsymbol{\Theta} = (\boldsymbol{\beta}_1, \boldsymbol{\beta}_2, \cdots, \boldsymbol{\beta}_m)^{\top}$, where $\boldsymbol{\beta}_i$ is the parameter vector corresponding to $i$-th arm (Bastani & Bayati, 2020).*

3. *(continuum arm bandit (without contextuals)) Suppose the arm in original continuum arm bandit is denoted by $a$, and the mean reward function is $f(a)$. Since all continuous function on a bounded interval can be approximated by polynomial functions to arbitrary precision, it's reasonable to assume $f(a)$ to be polynomial of order $n$, which is not known precisely and only an upper bound $N$ is known. Let $\boldsymbol{a} = (1, a, a^2, a^3, \cdots, a^n, \cdots, a^N)$, and suppose the first element of $\boldsymbol{x}$ is constant 1, then $\boldsymbol{\Theta}_{i,j} = \frac{1}{i!} f^{(i)}(a)$ for $j = 1$ and $\boldsymbol{\Theta}_{i,j} = 0$ for $j \neq 1$. Clearly, $\boldsymbol{\Theta}$ is rank $n$.*

## 3 Hi-CCAB ALGORITHM AND THEORETICAL RESULTS

In this section, we present our learning algorithm with a regret upper bound. Specifically, we detail the Hi-CCAB algorithm in Section 3.1 and establish an upper bound for its convergence rate of the time-averaged expected cumulative regret in Section 3.2.

### 3.1 DESCRIPTION OF THE LEARNING ALGORITHM

Our policy consists of two phases for each period $t \in [T]$: the first phase learns a low-rank representation and the second phase determines the assortment and the selling prices. In the first phase, our policy estimates $\widehat{\boldsymbol{\Theta}}_t$ by an penalized least-square estimator using $(\boldsymbol{a}_i, \boldsymbol{x}_{i,j}, r_{i,j})$ for $i = 1, \ldots, t$ and $l = 1, \ldots, L$. Based on $\widehat{\boldsymbol{\Theta}}_t$, we look for the optimal assortment and pricing within the action space $\mathcal{A}_t$. Algorithm 1 describes the detailed procedure of our policy.

**Low-rank representation learning.** As mentioned in Section 2, both the arm and the contextual vectors $\boldsymbol{a} \in \mathbb{R}^{d_a}$ and $\boldsymbol{x} \in \mathbb{R}^{d_x}$ are high-dimensional, and thus $\boldsymbol{\Theta} \in \mathbb{R}^{d_a \times d_x}$ is also high-dimensional. Fortunately, there often exists structure in both the arm and covariate space as explained in Section 1. To leverage the underlying structure, we impose a low-rank assumption on $\boldsymbol{\Theta}$, which automatically explores the effect of the low-rank structure and the relationships between the action and the contextual arms.

To estimate the low-rank representation of $\boldsymbol{\Theta}$ at time $t$, one can adopt the rank-penalized least square:

$$\widehat{\boldsymbol{\Theta}}_t := \arg\min_{\boldsymbol{\Theta}} \sum_{i=1}^{t} \sum_{j=1}^{L} \left( \boldsymbol{a}_i^{\top} \boldsymbol{\Theta} \boldsymbol{x}_{i,j} - r_{i,j} \right)^2 + \lambda_t \cdot \mathrm{rank}(\boldsymbol{\Theta}) \tag{4}$$

where $\lambda_t > 0$ is the penalization parameter and $\mathrm{rank}(\boldsymbol{\Theta})$ is the rank of the matrix $\boldsymbol{\Theta}$. However, the rank penalization makes (4) a non-convex problem, leading to computational challenges. To address the computational challenges, the rank penalization term is often replaced by the nuclear norm in matrix estimation and completion literature so that the optimization problem becomes a convex problem. We adopt a similar idea and our objective function then becomes:

$$\widehat{\boldsymbol{\Theta}}_t := \arg\min_{\boldsymbol{\Theta}} \sum_{i=1}^{t} \sum_{j=1}^{L} \left( \boldsymbol{a}_i^{\top} \boldsymbol{\Theta} \boldsymbol{x}_{i,j} - r_{i,j} \right)^2 + \lambda_t \cdot \|\boldsymbol{\Theta}\|_*. \tag{5}$$

The penalization parameter $\lambda_t$ is updated in each iteration such that $\lambda_t = \lambda_0/\sqrt{t}$ where $\lambda_0$ is the initialized penalization parameter, which can be chosen by cross-validation or guided by $\|\frac{1}{2t_1 L}\sum_{i=1}^{t_1}\sum_{j=1}^{L}|\boldsymbol{a}_i^\top \widehat{\boldsymbol{\Theta}}_{t_1}\boldsymbol{x}_{i,j} - r_{i,j}|\boldsymbol{x}_{i,j}\boldsymbol{a}_i^\top\|_2$.

---

**Algorithm 1:** The `Hi-CCAB` Algorithm.

---

**Result:** Actions $\boldsymbol{a}_{t_1+1}, \ldots, \boldsymbol{a}_T$.

**Input:** The number of steps for initialization $t_1$, set of possible actions $\mathcal{A}_{t_1}$, action vectors based on domain knowledge $\{\boldsymbol{a}_i\}_{i=1}^{t_1}$, covariates vector $\{\boldsymbol{x}_{i,j}\}_{i=1}^{t_1}$, rewards $r_{i,j}$ for $j = 1, \ldots, L$, and exploration parameter $h$;

**Initialization:** $\lambda_0 \leftarrow \|\frac{1}{2t_1 L}\sum_{i=1}^{t_1}\sum_{j=1}^{L}|\mathbf{a}_i^\top\widehat{\boldsymbol{\Theta}}_{t_1}\mathbf{x}_{i,j} - r_{i,j}|\mathbf{x}_{i,j}\mathbf{a}_i^\top\|_2$, $t \leftarrow t_1$;

**while** $t < T$ **do**

$\quad \lambda_t \leftarrow \lambda_0/\sqrt{t}$;

$\quad$**Low-rank representation learning:**

$\quad\quad \widehat{\boldsymbol{\Theta}}_t \leftarrow \arg\min_{\boldsymbol{\Theta}} \frac{1}{tL}\sum_{i=1}^{t}\sum_{j=1}^{L}(\boldsymbol{a}_i^\top\boldsymbol{\Theta}\boldsymbol{x}_{i,j} - r_{i,j})^2 + \lambda_t\|\boldsymbol{\Theta}\|_*$;

$\quad$**Policy learning:**

$\quad\quad \hat{\boldsymbol{a}}_{t+1} \leftarrow \arg\max_{\boldsymbol{a}\in\mathcal{A}_t}\sum_{j=1}^{L}\boldsymbol{a}^\top\widehat{\boldsymbol{\Theta}}_t\boldsymbol{x}_{t+1,j}$;

$\quad\quad$*Exploitation if* $t \notin \{\lfloor w^{\frac{3}{2}}\rfloor : w \in \mathbb{Z}_+\}$: $\boldsymbol{a}_{t+1} \leftarrow \hat{\boldsymbol{a}}_{t+1}$;

$\quad\quad$*Exploration if* $t \in \{\lfloor w^{\frac{3}{2}}\rfloor : w \in \mathbb{Z}_+\}$: $\boldsymbol{a}_{t+1} \leftarrow \hat{\boldsymbol{a}}_{t+1} + \boldsymbol{\delta}_{t+1}$, update action space $\mathcal{A}_{t+1}$

$\quad\quad\quad$ 1. $\boldsymbol{\delta}_{t+1} \sim N(\mathbf{0}_{d_a}, h\boldsymbol{I}_{d_a})$ or;

$\quad\quad\quad$ 2. $\boldsymbol{\delta}_{t+1} \sim N(\mathbf{0}_{d_a}, diag(\hat{\boldsymbol{\tau}}_t^2))$ and $\hat{\boldsymbol{\tau}}_{t,j}^2 = sd(\{\tilde{\boldsymbol{a}}_{i,j}\}_{i=1}^{t})$, $sd(\cdot)$ calculates the standard error;

$\quad$Apply action $\boldsymbol{a}_{t+1}$ and observe reward $r_{t+1,j}$ for $j = 1, \ldots, L$;

$\quad t \leftarrow t+1$;

**end**

---

**Policy learning.** Once we estimated the low-rank representation of $\boldsymbol{\Theta}$, we can proceed to the action step. The goal of the action step is to *exploit* the knowledge we learned from the previous time, i.e., $\widehat{\boldsymbol{\Theta}}_t$, so as to decide on the next action $\boldsymbol{a}_{t+1}$ that maximizes the reward, and at the same time to *explore* actions that better inform the true $\boldsymbol{\Theta}$, which in turns will help make better decision to achieve higher long-term rewards. Specifically, given $\widehat{\boldsymbol{\Theta}}_t$ and the covariate $\boldsymbol{x}_{t+1,j}$ for $j = 1, \ldots, L$, we look for an action $\hat{\boldsymbol{a}}_{t+1}$ in the action space $\mathcal{A}_t$ that maximizes the total rewards across $L$ objects:

$$\hat{\boldsymbol{a}}_{t+1} := \arg\max_{\boldsymbol{a}\in\mathcal{A}_t}\sum_{j=1}^{L}\boldsymbol{a}^\top\widehat{\boldsymbol{\Theta}}_t\boldsymbol{x}_{t+1,j}. \tag{6}$$

We further perturb $\hat{\boldsymbol{a}}_{t+1}$ for the purpose of exploration by adding random noise to each coordinate when $t \in \{\lfloor w^{\frac{3}{2}}\rfloor : w \in \mathbb{Z}_+\}$, i.e., $\boldsymbol{a}_{t+1} = \hat{\boldsymbol{a}}_{t+1} + \boldsymbol{\delta}_{t+1}$ where $\boldsymbol{\delta}_{t+1} \sim N(\mathbf{0}_{d_a}, h\boldsymbol{I}_{d_a})$ and $h$ is a tuning parameter. The intuition for $\lfloor w^{\frac{3}{2}}\rfloor$ is to explore more in the initial stage and exploit more in the later stage of the algorithm. To be specific, there are around $T^{\frac{2}{3}}$ steps for exploration before time $T$. The density of exploration at a small time frame around $T$ is $T^{-\frac{1}{3}}$, which goes to zero as $T \to \infty$. Note that the exponent can be any number larger than 1, instead of $\frac{3}{2}$, which will affect the convergence rate of the regret as we will discuss later in Remark 2. The polynomial form can be changed as well. For each exploration step, one can also let $\boldsymbol{\delta}_{t+1} \sim N(\mathbf{0}_{d_a}, diag(\hat{\boldsymbol{\tau}}_t))$ where each element of $\hat{\boldsymbol{\tau}}_t$ is the coordinate-wise standard error of the previous actions $\{\boldsymbol{a}_i\}_{i=1}^{t}$. The intuition is to avoid tuning parameter $h$ while taking the right scale. Finally we update the action space $\mathcal{A}_{t+1}$ according to $\boldsymbol{a}_{t+1}$. For example, if the action space $\mathcal{A}_t \in \mathbb{R}^{d_a}$ can be defined by an upper limit $\bar{\boldsymbol{a}}_t$ and a lower limit $\underline{\boldsymbol{a}}_t$, then we simply expand the action space by pushing the boundary of each coordinate to $\boldsymbol{a}_{t+1,j}$ if $\boldsymbol{a}_{t+1,j} \notin [\underline{\boldsymbol{a}}_{t,j}, \bar{\boldsymbol{a}}_{t,j}]$ for $j = 1, \ldots, d_a$.

**Remark 1.** *To take advantage of the interpretability of our model, we can further explore the structure of the $\widehat{\boldsymbol{\Theta}}_t$. Specifically, we can apply singular value decomposition (SVD) on $\widehat{\boldsymbol{\Theta}}_t$ to explore the underlying latent structure of the covariates from the right singular vectors; and apply SVD on $(\widehat{\boldsymbol{\Theta}}_t \sum_{j}^{L}\boldsymbol{x}_{t,j})$ to explore the latent structure of the arms from the left singular vectors. One can further rotate the singular vectors so as to reveal the underlying factors using techniques in factor analysis such as Varimax (Kaiser, 1958; Rohe & Zeng, 2020) or to perform clustering analysis by performing $K$-means on the singular vectors.*

## 3.2 THEORETICAL RESULTS

In this section, we establish in Theorem 1 that the convergence rate of time-averaged expected cumulative regret for Algorithm 1 is at least as fast as $T^{-\frac{2}{15}}$ and outline the proof strategy for Theorem 1. We consider the time-averaged expected cumulative regret since it measures the trend of the newly incurred regret, in the long run, more directly; on the other hand, the expected cumulative regret grows with time $T$, which is less interpretable. This theorem implies that the newly incurred regret, roughly speaking, converges to zero fast.

**Theorem 1.** *Suppose $\boldsymbol{x}_{t,l} \overset{i.i.d}{\sim} N(\boldsymbol{0}_{d_x}, \boldsymbol{I}_{d_x})$, and the errors $(\varepsilon_{t,j})$ defined in reward model (1) follows normal distribution: $\varepsilon_{t,j} \overset{i.i.d}{\sim} N(0, \sigma^2)$. Suppose $\boldsymbol{\Theta}$ is rank $d$. Suppose the exploration step in Algorithm 1 is $\boldsymbol{a}_t = \hat{\boldsymbol{a}}_t + \boldsymbol{\delta}_t$ for $t \in \{\lfloor w^{\frac{3}{2}} \rfloor : w \in \mathbb{Z}_+\}$ where $\boldsymbol{\delta}_t \sim N(\boldsymbol{0}_{d_a}, h\boldsymbol{I}_{d_a})$, $\mathcal{A}_t = \{\boldsymbol{a} \in \mathbb{R}^{d_a} : \|\boldsymbol{a}\| \leq 1\}$, then there is a $T_1$ such that for $T \geq T_1$, the expected cumulative regret of the Algorithm 1, $R_T^\pi$, satisfies*

$$\frac{R_T^\pi}{T} \leq 2\sqrt{Ld_x}\|\boldsymbol{\Theta}\|_2 T_1 T^{-1} + \frac{72}{5}\lambda_0 \frac{\sqrt{2dd_xL}}{h^2}T^{-\frac{1}{6}} + \frac{60}{13}\sqrt{Ld_x}\|\boldsymbol{\Theta}\|_2 T^{-\frac{2}{15}} + \frac{90}{13}\frac{\sigma(d_x+1)}{h^2}T^{-\frac{2}{15}},$$

$$(7)$$

*where $T_1 = C_{h,L,\lambda_0}(d_x + d_a)^6 (log(d_x + d_a))^3$ and the constant $C_{h,L,\lambda_0}$ depends on $h$, $L$ and $\lambda_0$. For $T \leq T_1$, $\frac{R_T^\pi}{T} \leq 2\sqrt{Ld_x}\|\boldsymbol{\Theta}\|_2$.*

**Remark 2** (Convergence rate). *An intuitive understanding of Theorem 1 is that the expected regret incurred each time converges to zero at a speed at least $T^{-\frac{2}{15}}$ as $T$ going to infinity. The convergence rate depends on the frequency of the exploration which depends on the exponent $\frac{3}{2}$ in the exploration set, $\{\lfloor w^{\frac{3}{2}} \rfloor : w \in \mathbb{Z}_+\}$. Recall that the exponent can be changed with any number larger than 1, which can be considered as a tuning parameter.*

**Remark 3** ("Burnout" term). *The first term in inequality (7) is a "burnout" term, where the algorithm is gaining knowledge of $\boldsymbol{\Theta}$ from scratch. We do not impose any assumptions on these starting steps so that we have a relative conservative "burnout" term. However, in practice, we usually have historical data to start with so that the algorithm can start from a reasonable estimation of $\boldsymbol{\Theta}$ and much smaller "burnout" term. Recall that the exponent of the exploration set can be any number larger than 1. The order of the "burnout" term depends on the exponent of the $w$ in the exploration set — the more exploration there is, the smaller the "burnout" term. The exponent can be chosen depending on the situation — how ample the historical data is.*

**Remark 4** (Constant $C_{h,L,\lambda_0}$ of $T_1$). *While constant $C_{h,L,\lambda_0}$ depends on $h, L, \lambda_0$, the primary dependency is actually on $h$ and $L$. The order of $\lambda_0$ in terms of dimensions and noise level is $\sigma\sqrt{d_x}$. We do not assume the order of $\lambda_0$ or bound it with a high probability bound in order to show its role in time-averaged expected cumulative regret. If we utilize the order $\sigma\sqrt{d_x}$, then $C_{h,L,\lambda_0}$ can be replaced by a constant depending on $h$ and $L$ only.*

**Remark 5** (Dependence on dimensions $d_a, d_x$ and rank $d$). *When $T$ is small, the "burnout" term (the first term) dominates. It depends on $T$ and the dimensions but not the rank as $(d_a + d_x)^6 (log(d_a + d_x))^3 T^{-1}$, whose order depends on the exponent defining the exploration set (i.e., how frequent we explore). As $T$ grows, the second term dominates. Recall Remark 4, $\lambda_0$ is of order $\sigma\sqrt{d_x}$, so the second terms depends on $T, d_x$ and $d$ but not $d_a$ at the order of $\Omega(d_x\sqrt{d}T^{-\frac{1}{6}})$. Without the low-rank assumption, the order would be $\Omega(d_x^{\frac{3}{2}}T^{-\frac{1}{6}})$ instead. When $T$ becomes even larger, the last two terms dominates, at the order $\Omega(d_x T^{-\frac{2}{15}})$. However, the last case rarely happens, as it requires the order of $T$ equal to or larger than $d^{15}$. Therefore, taking dimensions and rank into consideration, the time-averaged expected cumulative regret is mostly at the order of $\Omega(d_x\sqrt{d}T^{-\frac{1}{6}})$.*

**Proof sketch**  We outline the proof strategy for Theorem 1. There are two major steps: (1) bounding the estimation error for the low-rank representation matrix estimator; (2) bounding the expected cumulative regret. The detailed proof of Theorem 1 is in Appendix A.

(1) Bounding the estimation error of $\widehat{\boldsymbol{\Theta}}_t$ with a high probability bound. Denote $\delta\boldsymbol{\Theta}_t = \widehat{\boldsymbol{\Theta}}_t - \boldsymbol{\Theta}$. We show that for a large $t$,

$$P\left(\|\delta\boldsymbol{\Theta}_t\|_F \leq \frac{3}{T^{\frac{2}{15}}}\frac{\sigma\sqrt{d_x+1}}{\sqrt{L}h^2} + 6\lambda_0\frac{\sqrt{2d}}{h^2T^{\frac{1}{6}}}\right) \leq 1 - \left(\frac{3}{t} + \frac{2}{t^2} + \frac{2}{Lt} + \frac{2}{L^3t^3} + \frac{1}{t^{\frac{2}{15}}}\right).$$

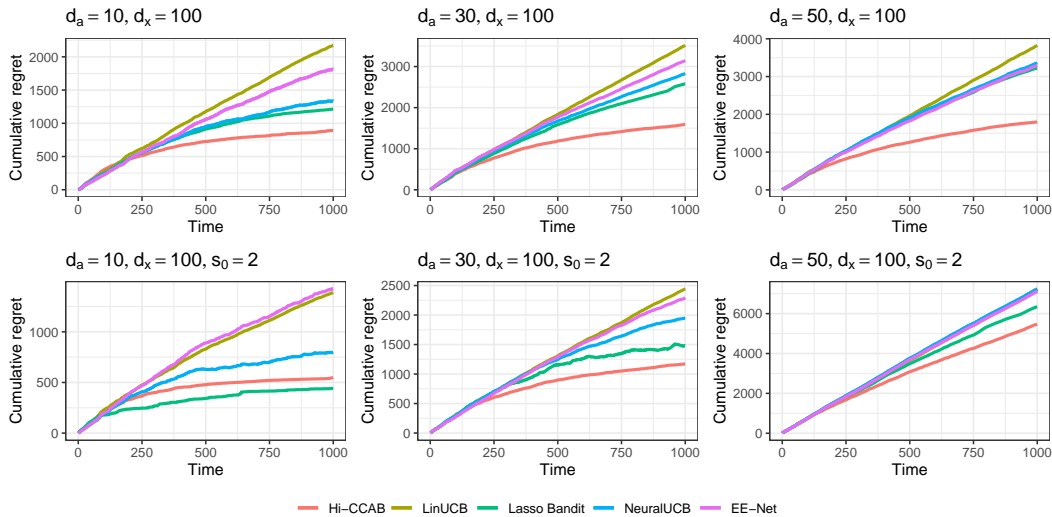

Figure 1: Cumulative regret under the non-sparse (first row) and sparse (second row) settings.

Note that the action taken is based on previous estimators and affect the accuracy of future estimators, leading to lots of dependencies. The classical matrix completion results can no longer apply. Through careful use of conditional expectations, martingales, and empirical process we separate out different sources of randomness (i.e., $\delta_1, \cdots, \delta_t, \boldsymbol{x}_{1,\cdot}, \cdots, \boldsymbol{x}_{t,\cdot}$) to derive the bounds. Lemma 1 establishes a restricted-strong-convexity-type result of the sum of squares in the objective function. Lemma 2 establishes a Lipschitz-type result of the sum of squares in the objective function. Further analysis of the nuclear-norm-penalized sum of squares with the two lemmas and low-rank properties gives the tail bound of the estimation error.

(2) Bounding the time-averaged expected regret. Let $Q_t = \{\|\delta\boldsymbol{\Theta}_t\|_F \leq \frac{3}{T^{\frac{2}{15}}}\frac{\sigma\sqrt{d_x+1}}{\sqrt{L}h^2} + 6\lambda_0\frac{\sqrt{2d}}{h^2 T^{\frac{1}{6}}}\}$ be the event such that $\delta\boldsymbol{\Theta}_t$ is bounded. We know that from the first step, for large $t$, $P(Q_t^c) \leq \frac{3}{t} + \frac{2}{t^2} + \frac{2}{Lt} + \frac{1}{L^3 t^3} + \frac{1}{t^{\frac{2}{15}}}$. Consider the expectation of the regret on $Q_t$ and $Q_t^c$ separately and both terms vanish with $t$ at the polynomial rate.

## 4 SIMULATION STUDY AND ASSORTMENT-PRICING CASE STUDY

In this section, we conduct simulation studies to compare the proposed `Hi-CCAB` with LinUCB (Li et al., 2010), Lasso Bandit (Bastani & Bayati, 2020), NeuralUCB (Zhou et al., 2020) and EE-Net (Ban et al., 2022); we then study the joint assortment-pricing problem on the e-commerce platform for one of the largest instant noodles producers in China. Details on the tuning parameters of each algorithms and additional results of the case study are provided in Appendix B-C.

**Simulation study** We consider the multi-armed linear bandit setup, a special case of our model as shown in Proposition 1, i.e., $\boldsymbol{\Theta} = (\boldsymbol{\beta}_1, \boldsymbol{\beta}_2, \cdots, \boldsymbol{\beta}_m)^\top$ so that each row of $\boldsymbol{\Theta}$ is the parameter of each arm for the multi-arm contextual bandit. Specifically, we set the number of arms $d_a = \{10, 30, 50\}$ and the dimension of covariates $d_x = 100$. For $\boldsymbol{\Theta}$, we consider a non-sparse and sparse case. For the non-sparse case, we generate $\boldsymbol{\Theta} = \boldsymbol{U}\boldsymbol{D}\boldsymbol{V}^\top$ where $\boldsymbol{U} \in \mathbb{R}^{d_a \times r}, \boldsymbol{V} \in \mathbb{R}^{d_x \times r}$ ($r = 5$), and $\boldsymbol{D}$ is a diagonal matrix with $(1, .9, .9, .8, .5)$ as the diagonal entries. All entries of $\boldsymbol{U}$ and $\boldsymbol{V}$ are first generated from i.i.d. $N(0, 1)$, and then applied Gram–Schmidt to make each column orthogonal. $\boldsymbol{U}$ is scaled to have length $\sqrt{d_a}$ so that the rewards are comparable across different $d_a$'s. For the sparse case, each row of $\boldsymbol{\Theta}$ are set as zero except for $s_0 = 2$ randomly selected elements that are drawn from $N(0, 1)$. We generate the covariate $\boldsymbol{x} \overset{i.i.d}{\sim} N(0, \boldsymbol{I}_{d_x})$ and the rewards from (1) with $\sigma = 0.1$.

Figure 1 shows the cumulative regret (averaged over 50 simulations). For the non-sparse case, `Hi-CCAB` converges faster than all other methods. The advantage of `Hi-CCAB` is more pronounced when the dimension of the arms becomes larger. For the sparse case, which is not to the advantage of `Hi-CCAB`, when the dimension of arms is relatively small ($d_a = 10$), Lasso Bandit converges faster but the gap between `Hi-CCAB` and Lasso Bandit is small. As the number of arms increases, `Hi-CCAB` outperforms all other methods.

**Assortment-pricing case study.** The original data contains daily sales of 176 products across 369 cities from March 1st, 2021 to May 31st, 2022 ($T = 456$ days). We aggregate the sales by 31 provinces. Each product is of either single or assorted flavors (13 possible flavors) with different

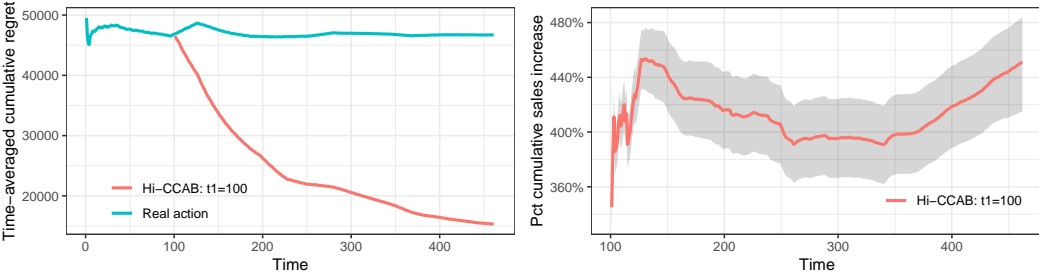

(a) Time-averaged cumulative regret.          (b) Percentage cumulative sales gain.

Figure 2: Performance of `Hi-CCAB` compared with real actions over 100 simulations. The boundaries of the shadow boundaries are the 5-th and 95-th quantiles.

counts. The assortment and price of each product changed daily. In addition, we know the dates for promotion. The assortment, prices, and promotions were the same across locations. The maximum number of products to be shown on the homepage is $K = 30$. The total possible combinations are then $\binom{176}{30}$ and therefore if we consider one combination as one arm, we are facing extremely high-dimensional arms, for which most multi-arm bandit algorithms are not applicable.

To apply `Hi-CCAB`, we specify the arms $\boldsymbol{a}_t$ and the covariate vectors $\{\boldsymbol{x}_{t,j}\}_{j=1}^{L=31}$ at given time $t$ following the setup in Example 1. The arm is represented as $\boldsymbol{a} = (\tilde{\boldsymbol{p}}_1, \tilde{\boldsymbol{p}}_1^2, p_1, p_1^2, promo_1, promo_1^2, \cdots, \tilde{\boldsymbol{p}}_K, \tilde{\boldsymbol{p}}_K^2, p_K, p_K^2, promo_K, promo_K^2, 1) \in \mathbb{R}^{2(m+2)K+1=901}$ where $\tilde{\boldsymbol{p}}_k = (\#flavor_{k,1}, \cdots, \#flavor_{k,m})$ is a vector of non-negative integers to denote the counts of $m = 13$ flavors, $p_k$ is the price, $promo_k$ is the indicator of promotion of product $k$, and $\tilde{\boldsymbol{p}}_k^2$ is the element-wise quadratics. The covariate $\boldsymbol{x}_{t,j} \in \mathbb{R}^{50}$ for location $j$ includes dummy variables of 31 provinces, year 2021/2022, 12 months, weekdays, and an indicator of annual sales event on Jun 18 and Nov 11. More details are deferred to Appendix C.

To run simulations using the dataset, we first create a pseudo-truth model. To be specific, we estimate $\boldsymbol{\Theta}$ and $\sigma$ using all data of 456 days and consider them as the pseudo ground truth. We perform a sanity check on our model assumption (1), the pseudo ground truth against our data before preceding to the formal analysis and further examine the structure of the representation matrix $\boldsymbol{\Theta}$ in Appendix C. We evaluate the performance of `Hi-CCAB` in terms of the cumulative regret (2) and the percentage gain of the cumulative sales by comparing with the original actions, since no existing bandit algorithm is applicable to this problem.

Figure 2a shows the time-averaged cumulative regret (averaged over 100 simulations) and Figure 2b shows the percentage gain in cumulative sales compared to the real sales The time-averaged cumulative regret of `Hi-CCAB` converges to zero while that of original actions remains flat. In terms of percentage gain in cumulative sales, `Hi-CCAB` boosts cumulative sales by more than 4 times. On a separate note, `Hi-CCAB` with exploration performs better in terms of both cumulative regret and percentage sales gain than `Hi-CCAB` without exploration.

## 5   CONCLUSION

With an increasing demand for online decision-making, the bandit problem is receiving increasingly more attention from both theoreticians and practitioners. Even though the volume of bandit literature has been expanding, there exists very little literature on high-dimensional continuum armed contextual bandit with high-dimensional covariates. In this work, we formulate and propose a model for this problem. Our model is general as it unifies a large class of bandit problems and has interpretability and predictive power. We propose an efficient algorithm `Hi-CCAB` by adopting the low-rank matrix estimator and provide an upper bound for its convergence rate in terms of the time-averaged expected cumulative regret. The generality and flexibility of our model allow for its application in the joint assortment-pricing problem, where the assortment and pricing optimization problems have been studied extensively in operation research separately but not their joint optimization problem. By applying our model and algorithm to the real case study on the joint assortment-pricing problem for one of the largest instant noodles producers in China, we are able to boost the sales by four times and provide insights into the underlying structure of the effect on the reward of the arms and covariates such as purchasing behaviors. Therefore, both the theoretical and the real case study indicate that our model and algorithm can be effective for the high-dimensional continuum armed and high-dimensional contextual bandit problem faced by decision-makers in various fields. Since our model is new to the bandit literature, there is space for improvement in our regret analysis. This is an interesting future direction.

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

## APPENDIX A   PROOF OF THEOREM 1

In this proof, we denote the true parameter as $\boldsymbol{\Theta}^*$.

Let $\mathcal{L}_T(\boldsymbol{\Theta}) := \frac{1}{2LT} \sum_{t=1}^{T} \sum_{l=1}^{L} (\boldsymbol{a}_{t,l}^{\top} \boldsymbol{\Theta} \boldsymbol{x}_{t,l} - r_{t,l})^2$. Then we have the following lemmas that we will prove later.

**Lemma 1.** *Suppose all the assumptions in Theorem 1 holds. Denote $\mathcal{E}_T(\Delta) = \mathcal{L}_T(\boldsymbol{\Theta}^* + \Delta) - \mathcal{L}_T(\boldsymbol{\Theta}^*) - \langle \nabla \mathcal{L}_T(\boldsymbol{\Theta}^*), \Delta \rangle$. Then with probability at least $1 - \frac{1}{LT} - \frac{2}{T} - \frac{1}{T^2}$,*

$$\mathcal{E}_T(\Delta) \geq \frac{\lfloor T^{\frac{2}{3}} \rfloor}{2T} h^2 \|\Delta\|_F^2 - 14 T^{-\frac{2}{3}} (h + h^2)(2d_x + 2d_a + 6\log T + 6\log L)^2 \log T \|\Delta\|_2^2. \quad (8)$$

**Lemma 2.** *Suppose all the assumptions in Theorem 1 holds. With probability at least $1 - \frac{1}{T^{\frac{2}{15}}} - \frac{2}{L^3 T^3} - \frac{1}{LT} - \frac{1}{T} - \frac{1}{T^2}$, the following holds for all $\Delta$*

$$
\begin{aligned}
|\langle \nabla \mathcal{L}_T(\boldsymbol{\Theta}^*), \Delta \rangle| \leq & \|\Delta\|_F \frac{\sigma \sqrt{d_x + 1}}{\sqrt{LT}} T^{\frac{1}{30}} + \\
& \left( 2h\sigma T^{-2/3} \log T \sqrt{\frac{\max\{d_a, d_x\} log(d_a + d_x)}{L}} + \right. \\
& \left. \frac{8h\sigma}{T} \sqrt{log(TL)} \sqrt{(d_x + 3log(LT))(d_a + 3logT)} (log(d_x + d_a) + 2logT) \right) \|\Delta\|_*.
\end{aligned}
\tag{9}
$$

Recall the definition of $\hat{\boldsymbol{\Theta}}_t$, we know that

$$\mathcal{L}_T(\hat{\boldsymbol{\Theta}}_T) + \lambda_T \|\hat{\boldsymbol{\Theta}}_T\|_* \leq \mathcal{L}_T(\boldsymbol{\Theta}^*) + \lambda_T \|\boldsymbol{\Theta}^*\|_*. \tag{10}$$

Denote $\delta\boldsymbol{\Theta}_t = \hat{\boldsymbol{\Theta}}_t - \boldsymbol{\Theta}^*$ and for notation simplicity we will drop the subscript $t$ for $\delta\boldsymbol{\Theta}_t$ in the following when there is no confusion. Equation (10) then implies that

$$\mathcal{E}_T(\delta\boldsymbol{\Theta}) \leq -\langle \nabla \mathcal{L}_T(\boldsymbol{\Theta}^*), \delta\boldsymbol{\Theta} \rangle + \lambda_T (\|\boldsymbol{\Theta}^*\|_* - \|\boldsymbol{\Theta}^* + \delta\boldsymbol{\Theta}\|_*). \tag{11}$$

Suppose the singular value decomposition of $\boldsymbol{\Theta}^*$ is $\boldsymbol{\Theta}^* = \boldsymbol{U}\boldsymbol{S}\boldsymbol{V}^{\top}$, where $\boldsymbol{S}$ is an $d \times d$ diagonal matrix. Let $\boldsymbol{U}_{\top}$ be an $d_a \times (d_a - d)$ matrix satisfying $(\boldsymbol{U}, \boldsymbol{U}_{\perp})(\boldsymbol{U}, \boldsymbol{U}_{\perp})^{\top} = \boldsymbol{I}_{d_a}$. We define $\boldsymbol{V}_{\perp}$ similarly.

Denote $\delta\boldsymbol{\Theta}_\perp = \boldsymbol{U}_\perp^\top \delta\boldsymbol{\Theta}\boldsymbol{V}_\perp$. Then $\|\boldsymbol{\Theta}^* + \delta\boldsymbol{\Theta}\|_* \geq \|\boldsymbol{\Theta}^* + \delta\boldsymbol{\Theta}_\perp\|_* - \|\delta\boldsymbol{\Theta} - \delta\boldsymbol{\Theta}_\perp\|_* = \|\boldsymbol{\Theta}^*\|_* + \|\delta\boldsymbol{\Theta}_\perp\|_* - \|\delta\boldsymbol{\Theta} - \delta\boldsymbol{\Theta}_\perp\|_* \geq \|\boldsymbol{\Theta}^*\|_* + \|\delta\boldsymbol{\Theta}_\perp\|_* - \sqrt{2d}\|\delta\boldsymbol{\Theta} - \delta\boldsymbol{\Theta}_\perp\|_F$.

Going back to Inequality (11), and combing with Lemma 1 and Lemma 2, we have, with probability at least $1 - \frac{3}{T} - \frac{2}{T^2} - \frac{2}{LT} - \frac{2}{L^3 T^3} - \frac{1}{T^{\frac{1}{3}}}$, the following holds

$$
\left( \frac{\lfloor T^{\frac{2}{3}} \rfloor}{2T} h^2 - 14 T^{-\frac{2}{3}} (h + h^2) \left( 2d_x + 2d_a + 6\log T + 6\log L \right)^2 \log T \right) \|\delta\boldsymbol{\Theta}\|_F^2
$$

$$
\leq \|\delta\boldsymbol{\Theta}\|_F \frac{\sigma\sqrt{d_x + 1}}{\sqrt{LT}} T^{\frac{1}{30}} +
$$

$$
\left( \frac{8h\sigma}{T} \sqrt{log(TL)} \sqrt{(d_x + 3log(LT))(d_a + 3logT)}(log(d_x + d_a) + 2logT) + \right. \tag{12}
$$

$$
\left. 2h\sigma T^{-2/3} logT \sqrt{\frac{\max\{d_a, d_x\} log(d_a + d_x)}{L}} \right) \left( \|\delta\boldsymbol{\Theta} - \delta\boldsymbol{\Theta}_\perp\|_* + \|\delta\boldsymbol{\Theta}_\perp\|_* \right)
$$

$$
+ \lambda_0 \frac{\sqrt{T}}{T} \sqrt{2d} \|\delta\boldsymbol{\Theta}\|_F - \lambda_0 \frac{\sqrt{T}}{T} \|\delta\boldsymbol{\Theta}_\perp\|_*.
$$

Note that $\|\delta\boldsymbol{\Theta} - \delta\boldsymbol{\Theta}_\perp\|_* \leq \sqrt{2d}\|\delta\boldsymbol{\Theta} - \delta\boldsymbol{\Theta}_\perp\|_F$, divide both side with $\|\delta\boldsymbol{\Theta}\|_F$ and multiply both sides with $3T^{\frac{1}{3}}/h^2$. Suppose $T_1$ satisfies

$$
T_1 \geq 8,
$$

$$
T_1^{\frac{1}{3}} \geq 12 \times 14 (1 + \frac{1}{h}) \left( 2d_x + 2d_a + 6\log T_1 + 6\log L \right)^2 \log T_1,
$$

$$
\lambda_0 T_1^{\frac{1}{6}} \geq \frac{8h\sigma}{T_1^{\frac{1}{3}}} \sqrt{\log(T_1 L)} \sqrt{(d_x + 3\log(LT))(d_a + 3\log T_1)}(\log(d_x + d_a) + 2logT_1) + \tag{13}
$$

$$
2h\sigma T_1^{-2/3} \log T_1 \sqrt{\frac{\max\{d_a, d_x\} \log(d_a + d_x)}{L}}
$$

we have

$$
\|\delta\boldsymbol{\Theta}\|_F \leq \frac{3}{T^{\frac{2}{15}}} \frac{\sigma\sqrt{d_x + 1}}{\sqrt{L}h^2} + 6\lambda_0 \frac{\sqrt{2d}}{h^2 T^{\frac{1}{6}}}. \tag{14}
$$

Note that there is a constant $C_{h,L,\lambda_0}$ depending on L, h and $\lambda_0$ such that for

$$
T_1 \geq C_{h,L,\lambda_0} (d_x + d_a)^6 \left( log(d_x + d_a) \right)^3,
$$

Inequalities (13) holds.

Next we will proceed to bound the regret. Denote the event that (14) holds to be $Q_t$ and its complement as $Q_t^c$. Then $\mathbb{P}(Q_t^c) \leq \frac{3}{T} + \frac{2}{T^2} + \frac{2}{LT} + \frac{2}{L^3 T^3} + \frac{1}{T^{\frac{2}{15}}}$ and $\left( Q_t^c, \hat{\boldsymbol{\Theta}}_t \right) \perp\!\!\!\perp \boldsymbol{b}_{t+1}$. Let the oracle

optimal action at time $t$ be $\boldsymbol{a}_t^*$ and $\boldsymbol{b}_t = \sum_{l=1}^{L} \boldsymbol{x}_{t,l}$. Then

$$
R_T^\pi - \mathbb{E}\left(\sum_{t=0}^{T_1-1}\sum_{l=1}^{L}\left(\boldsymbol{a}_{t+1}^{*\top}\boldsymbol{\Theta}^*\boldsymbol{x}_{t+1,l} - \boldsymbol{a}_{t+1}^\top\boldsymbol{\Theta}^*\boldsymbol{x}_{t+1,l}\right)\right)
$$

$$
\leq \mathbb{E}\left(\sum_{t=T_1}^{T-1}\sum_{l=1}^{L}\boldsymbol{a}_{t+1}^{*\top}\boldsymbol{\Theta}^*\boldsymbol{x}_{t+1,l} - \boldsymbol{a}_{t+1}^\top\boldsymbol{\Theta}^*\boldsymbol{x}_{t+1,l}\right)
$$

$$
\leq \mathbb{E}\left(\sum_{t=T_1}^{T-1}\left\langle \frac{\boldsymbol{\Theta}^*\boldsymbol{b}_{t+1}}{\|\boldsymbol{\Theta}^*\boldsymbol{b}_{t+1}\|_2} - \frac{\hat{\boldsymbol{\Theta}}_t\boldsymbol{b}_{t+1}}{\|\hat{\boldsymbol{\Theta}}_t\boldsymbol{b}_{t+1}\|_2}, \boldsymbol{\Theta}^*\boldsymbol{b}_{t+1}\right\rangle\right)
$$

$$
= \sum_{t=T_1}^{T-1}\mathbb{E}\left(\left\langle \frac{\boldsymbol{\Theta}^*\boldsymbol{b}_{t+1}}{\|\boldsymbol{\Theta}^*\boldsymbol{b}_{t+1}\|_2} - \frac{\hat{\boldsymbol{\Theta}}_t\boldsymbol{b}_{t+1}}{\|\hat{\boldsymbol{\Theta}}_t\boldsymbol{b}_{t+1}\|_2}, \boldsymbol{\Theta}^*\boldsymbol{b}_{t+1}\right\rangle \mathbb{1}\{Q_t\}\right) + \mathbb{E}\left(\left\langle \frac{\boldsymbol{\Theta}^*\boldsymbol{b}_{t+1}}{\|\boldsymbol{\Theta}^*\boldsymbol{b}_{t+1}\|_2} - \frac{\hat{\boldsymbol{\Theta}}_t\boldsymbol{b}_{t+1}}{\|\hat{\boldsymbol{\Theta}}_t\boldsymbol{b}_{t+1}\|_2}, \boldsymbol{\Theta}^*\boldsymbol{b}_{t+1}\right\rangle \mathbb{1}\{Q_t^c\}\right)
$$

$$
\leq \sum_{t=T_1}^{T-1}\left(\mathbb{E}\left(\left\langle \frac{(\boldsymbol{\Theta}^*-\hat{\boldsymbol{\Theta}}_t)\boldsymbol{b}_t}{\|\boldsymbol{\Theta}^*\boldsymbol{b}_{t+1}\|_2} + \frac{\|\hat{\boldsymbol{\Theta}}_t\boldsymbol{b}_{t+1}\|_2 - \|\boldsymbol{\Theta}^*\boldsymbol{b}_{t+1}\|_2}{\|\hat{\boldsymbol{\Theta}}_t\boldsymbol{b}_{t+1}\|_2\|\boldsymbol{\Theta}^*\boldsymbol{b}_{t+1}\|_2}\hat{\boldsymbol{\Theta}}_t\boldsymbol{b}_{t+1}, \boldsymbol{\Theta}^*\boldsymbol{b}_{t+1}\right\rangle \mathbb{1}\{Q_t\}\right)\right.
$$

$$
\left. + 2\|\boldsymbol{\Theta}^*\|_2\sqrt{\mathbb{E}[\|\boldsymbol{b}_{t+1}\|^2]}\left(\frac{3}{t} + \frac{2}{t^2} + \frac{2}{Lt} + \frac{2}{L^3t^3} + \frac{1}{t^{\frac{1}{3}}}\right)\right)
$$

$$
\leq \sum_{t=T_1}^{T-1}\mathbb{E}\left[2\|\delta\boldsymbol{\Theta}\|_2\|\boldsymbol{b}_{t+1}\|\mathbb{1}\{Q_t\}\right] + 2\sqrt{Ld_x}\|\boldsymbol{\Theta}^*\|_2\left(\frac{3}{t} + \frac{2}{t^2} + \frac{2}{Lt} + \frac{2}{L^3t^3} + \frac{1}{t^{\frac{2}{15}}}\right)
$$

$$
\leq \sum_{t=T_1}^{T-1}\mathbb{E}\left[2\|\delta\boldsymbol{\Theta}\|_F\|\boldsymbol{b}_{t+1}\|\mathbb{1}\{Q_t\}\right] + 2\sqrt{Ld_x}\|\boldsymbol{\Theta}^*\|_2\left(\frac{3}{t} + \frac{2}{t^2} + \frac{2}{Lt} + \frac{2}{L^3t^3} + \frac{1}{t^{\frac{2}{15}}}\right)
$$

$$
\leq 2\sum_{t=T_1}^{T-1}\sqrt{Ld_x}\left(\frac{3}{t^{\frac{2}{15}}}\frac{\sigma\sqrt{d_x+1}}{\sqrt{L}h^2} + 6\lambda_0\frac{\sqrt{2d}}{h^2t^{\frac{1}{6}}}\right) + 2\sqrt{Ld_x}\|\boldsymbol{\Theta}^*\|_2\left(\frac{3}{t} + \frac{2}{t^2} + \frac{2}{Lt} + \frac{2}{L^3t^3} + \frac{1}{t^{\frac{2}{15}}}\right).
$$
$$(15)$$

Similar arguments also give

$$
\mathbb{E}\left(\sum_{t=0}^{T_1-1}\sum_{l=1}^{L}\mathbb{E}\left(\boldsymbol{a}_t^{*\top}\boldsymbol{\Theta}^*\boldsymbol{x}_{t,l} - \boldsymbol{a}_t^\top\boldsymbol{\Theta}^*\boldsymbol{x}_{t,l}\right)\right) \leq T_1 \times 2\sqrt{Ld_x}\|\boldsymbol{\Theta}^*\|_2. \tag{16}
$$

Therefore, for $T \geq T_1$,

$$
\frac{R_T^\pi}{T} \leq 2\sqrt{Ld_x}\|\boldsymbol{\Theta}^*\|_2T_1T^{-1} + \frac{60}{13}\sqrt{Ld_x}\|\boldsymbol{\Theta}^*\|_2T^{-\frac{2}{15}} + \frac{90}{13}\frac{\sigma(d_x+1)}{h^2}T^{-\frac{2}{15}} + \frac{72}{5}\lambda_0\frac{\sqrt{2dd_xL}}{h^2}T^{-\frac{1}{6}}
$$
$$(17)$$

### A.1  PROOF OF LEMMA 2

Suppose $r_{t,l} = a_{t,l}\boldsymbol{\Theta}^*\boldsymbol{x}_{t,l} + \sigma\varepsilon_{t,l}$, then

$$
\nabla\mathcal{L}_T(\boldsymbol{\Theta}^*) = \frac{\sigma}{LT}\sum_{t=1}^{T}\sum_{l=1}^{L}-\varepsilon_{t,l}\boldsymbol{x}_{t,l}\boldsymbol{a}_{t,l}^\top
$$

$$
= \frac{\sigma}{LT}\sum_{l=1}^{L}-\varepsilon_{1,l}\boldsymbol{x}_{1,l}\boldsymbol{a}_{1,l}^\top \tag{18}
$$

$$
+ \frac{\sigma}{LT}\sum_{t=2}^{T}\sum_{l=1}^{L}\left(-\varepsilon_{t,l}\boldsymbol{x}_{t,l}\hat{\boldsymbol{a}}_t^\top - \varepsilon_{t,l}\boldsymbol{x}_{t,l}\boldsymbol{\delta}_t^\top\right)
$$

Now we consider the terms in (18) separately. Let

$$S_2 = \frac{\sigma}{LT} \sum_{l=1}^{L} -\varepsilon_{1,l} \boldsymbol{x}_{1,l} \boldsymbol{a}_{1,l}^\top + \frac{\sigma}{LT} \sum_{t=2}^{T} \sum_{l=1}^{L} -\varepsilon_{t,l} \boldsymbol{x}_{t,l} \hat{\boldsymbol{a}}_t^\top.$$

$$S_3 = \frac{\sigma}{LT} \sum_{t=2}^{T} \sum_{l=1}^{L} -\varepsilon_{t,l} \boldsymbol{x}_{t,l} \boldsymbol{\delta}_t^\top \tag{19}$$

Elementary Calculation show that

$$\mathbb{E}(\|S_2\|_F^4) \le \frac{\sigma^4(d_x^2 + 2d_x)}{L^2 T^2}. \tag{20}$$

Therefore,

$$P(\|S_2\|_F \ge \frac{\sigma\sqrt{d_x+1}}{\sqrt{LT}} T^{\frac{1}{30}}) \le \frac{1}{T^{\frac{2}{15}}}. \tag{21}$$

For $S_3$, let $G$ be an event defined as

$$\begin{aligned} G = \Big\{ & \max\{|\varepsilon_{t,l}| : 1 \le t \le T, 1 \le l \le L\} \le 3\sqrt{\log TL}, \\ & \max\{\|\boldsymbol{x}_{t,l}\|^2 : 1 \le t \le T, 1 \le l \le L\} \le 2d_x + 6\log LT, \\ & \max\{\|\boldsymbol{\delta}_t/h\|_2^2 : 1 \le t \le T\} \le 2d_a + 6\log T\Big\}. \end{aligned} \tag{22}$$

Then elementary calculation shows that

$$P(G^c) \le \frac{2}{T^3 L^3} + \frac{1}{LT} + \frac{1}{T}. \tag{23}$$

Using Matrix Bernstein Inequality (Tropp, 2012) on event G, we have the operator norm of $S_3$ on G is bounded as follows

$$P(\{\|\frac{LT}{\sigma} S_3\|_2 \ge \alpha\} \cap G) \le (d_x + d_a) \exp\Big(\frac{-\alpha^2}{2\sigma_{S_3}^2 + 2D\alpha/3}\Big), \tag{24}$$

where

$$\sigma_{S_3}^2 \ge \max\Big\{\Big\| \sum_{t=1}^{T} \mathbb{E}\Big( (\sum_{l=1}^{L} \varepsilon_{t,l} \boldsymbol{x}_{t,l} \boldsymbol{\delta}_t^\top)(\sum_{l=1}^{L} \varepsilon_{t,l} \boldsymbol{x}_{t,l} \boldsymbol{\delta}_t^\top)^\top \Big) \Big\|_2, \\ \Big\| \sum_{t=1}^{T} \mathbb{E}\Big( (\sum_{l=1}^{L} \varepsilon_{t,l} \boldsymbol{x}_{t,l} \boldsymbol{\delta}_t^\top)^\top (\sum_{l=1}^{L} \varepsilon_{t,l} \boldsymbol{x}_{t,l} \boldsymbol{\delta}_t^\top) \Big) \Big\|_2\Big\}, \tag{25}$$

and

$$D = \max_t \sup_{\text{event } G \text{ holds}} \|\sum_{l=1}^{L} -\varepsilon_{t,l} \boldsymbol{x}_{t,l} \boldsymbol{\delta}_t^\top\|_2 \le 6L\sqrt{\log TL} h\sqrt{(d_x + 3\log LT)(d_a + 3\log T)}. \tag{26}$$

Elementary calculation shows that taking

$$\sigma_{S_3}^2 = h^2 \lfloor T^{\frac{2}{3}} \rfloor L \max\{d_a, d_x\} \tag{27}$$

satisfies Equation (25).

Taking

$$\begin{aligned} \alpha = & 2hT^{\frac{1}{3}} \log T \sqrt{L \max\{d_a, d_x\} \log(d_a + d_x)} + \\ & 8hL\sqrt{\log TL}\sqrt{(d_x + 3\log(LT))(d_a + 3\log T)}(\log(d_x + d_a) + 2\log T) \end{aligned} \tag{28}$$

$$P(\{\|\frac{LT}{\sigma} S_3\|_2 \ge \alpha\} \cap G) \le \frac{1}{T^2}. \tag{29}$$

Therefore, we have

$$
\begin{aligned}
P\Big(\|S_3\|_2 &\le 2h\sigma T^{-2/3}logT\sqrt{\frac{\max\{d_a, d_x\}log(d_a + d_x)}{L}} + \\
&\frac{8h\sigma}{T}\sqrt{log(TL)}\sqrt{(d_x + 3log(LT))(d_a + 3logT)}(log(d_x + d_a) + 2logT)\Big) \\
&\ge 1 - \frac{2}{L^3 T^3} - \frac{1}{LT} - \frac{1}{T} - \frac{1}{T^2}
\end{aligned}
\tag{30}
$$

Recalling that

$$
|\langle \nabla \mathcal{L}_T(\Theta^*), \Delta \rangle| = |\langle S_2, \Delta \rangle + \langle S_3, \Delta \rangle| \le \|S_2\|_F \|\Delta\|_F + \|S_3\|_2 \|\Delta\|_*,
\tag{31}
$$

we get the statement of the lemma.

### A.2 PROOF OF LEMMA 1

Let $\boldsymbol{b}_t = \sum_{l=1}^{L} \boldsymbol{x}_{t,l}$. Let $\boldsymbol{\delta}_t = \boldsymbol{0}$ for exploitation rounds.

Then we know that

$$
\begin{aligned}
\mathcal{E}_T(\Delta) &= \frac{1}{2LT} \sum_{t=1}^{T} \sum_{l=1}^{L} (\boldsymbol{a}_{t,l}^\top \Delta \boldsymbol{x}_{t,l})^2 \\
&= \frac{1}{2LT} \sum_{t=1}^{T} \sum_{l=1}^{L} ((\frac{\boldsymbol{b}_t^\top \hat{\boldsymbol{\Theta}}_{t-1}^\top}{\|\boldsymbol{b}_t^\top \hat{\boldsymbol{\Theta}}_{t-1}^\top\|} + \boldsymbol{\delta}_t^\top) \Delta \boldsymbol{x}_{t,l})^2
\end{aligned}
\tag{32}
$$

Define

$$
\begin{aligned}
\mathcal{D}_T(\Delta) &= \frac{1}{2LT} \sum_{t=1}^{T} \sum_{l=1}^{L} \left( (\frac{\boldsymbol{b}_t^\top \hat{\boldsymbol{\Theta}}_{t-1}^\top}{\|\boldsymbol{b}_t^\top \hat{\boldsymbol{\Theta}}_{t-1}^\top\|} \Delta \boldsymbol{x}_{t,l})^2 + (\boldsymbol{\delta}_t^\top \Delta \boldsymbol{x}_{t,l})^2 \right), \\
\mathcal{D}_{1,T}(\Delta) &= \frac{1}{2LT} \sum_{t=1}^{T} \sum_{l=1}^{L} (\frac{\boldsymbol{b}_t^\top \hat{\boldsymbol{\Theta}}_{t-1}^\top}{\|\boldsymbol{b}_t^\top \hat{\boldsymbol{\Theta}}_{t-1}^\top\|} \Delta \boldsymbol{x}_{t,l})^2 \\
\mathcal{D}_{2,T}(\Delta) &= \frac{1}{2LT} \sum_{t=1}^{T} \sum_{l=1}^{L} (\boldsymbol{\delta}_t^\top \Delta \boldsymbol{x}_{t,l})^2
\end{aligned}
\tag{33}
$$

Then

$$
\mathcal{E}_T(\Delta) - \mathcal{D}_T(\Delta) = \frac{1}{LT} \sum_{t=1}^{T} \sum_{l=1}^{L} (\frac{\boldsymbol{b}_t^\top \hat{\boldsymbol{\Theta}}_{t-1}^\top}{\|\boldsymbol{b}_t^\top \hat{\boldsymbol{\Theta}}_{t-1}^\top\|} \Delta \boldsymbol{x}_{t,l})(\boldsymbol{\delta}_t^\top \Delta \boldsymbol{x}_{t,l})
\tag{34}
$$

Elementary calculation shows that

$$
\mathbb{E}(\mathcal{E}_T(\Delta) - \mathcal{D}_T(\Delta)) = 0,
\tag{35}
$$

and

$$
\mathbb{E}(\mathcal{D}_{2,T}(\Delta)) \ge \frac{\lfloor T^{\frac{2}{3}} \rfloor}{2T} h^2 \|\Delta\|_F^2.
\tag{36}
$$

Now we proceed with proving that the following two holds with high probability:

$$
\begin{aligned}
\inf_{\|\Delta\|_2 > 0} \frac{\mathcal{E}_T(\Delta) - \mathcal{D}_T(\Delta)}{\|\Delta\|_2^2} &\ge -7T^{-\frac{2}{3}}(h + h^2)(2d_x + 2d_a + 6\log T + 6\log L)^2 \log T \\
\inf_{\|\Delta\|_2 > 0} \frac{\mathcal{D}_{2,T}(\Delta) - \mathbb{E}(\mathcal{D}_{2,T}(\Delta))}{\|\Delta\|_2^2} &\ge -7T^{-\frac{2}{3}}(h + h^2)(2d_x + 2d_a + 6\log T + 6\log L)^2 \log T.
\end{aligned}
\tag{37}
$$

Note that $\|\boldsymbol{x}_{t,l}\|_2^2 \sim \chi_{d_x}^2$, $\|\boldsymbol{\delta}_t/h\|_2^2 \sim \chi_{d_a}^2$. Therefore, we have that

$$
\begin{aligned}
P(\sup_{t,l} \|\boldsymbol{x}_{t,l}\|_2^2 &\le d_x + 2\epsilon_1 + 2\sqrt{\epsilon_1 d_x}, \sup_t \|\boldsymbol{\delta}_t/h\|_2^2 \le d_a + 2\epsilon_2 + 2\sqrt{\epsilon_2 d_a}) \\
&\ge 1 - (LT \exp(-\epsilon_1) + T \exp(-\epsilon_2)).
\end{aligned}
\tag{38}
$$

Let $\epsilon_1 = 2 \log LT, \epsilon_2 = 2 \log T$.

Denote

$$U_1 = d_x + 2\epsilon_1 + 2\sqrt{\epsilon_1 d_x}, U_2 = d_a + 2\epsilon_2 + 2\sqrt{\epsilon_2 d_a}. \tag{39}$$

And let the event $O$ be

$$O = \{\sup_{t,l} \|\boldsymbol{x}_{t,l}\|_2^2 \leq U_1, \sup_t \|\boldsymbol{\delta}_t/h\|_2^2 \leq U_2\}. \tag{40}$$

For the following, we restrict our attention to event $O$.

Note that

$$\inf_{U_0/1.1\|\Delta\|_2 \leq U_0} \mathcal{E}_T(\Delta) - \mathcal{D}_T(\Delta) \geq \inf_{U_0/1.1\|\Delta\|_2 \leq U_0, \hat{\Theta}_{t-1}^\top \neq \boldsymbol{0} \text{ for } 1 \leq t \leq T} \mathcal{E}_T(\Delta) - \mathcal{D}_T(\Delta), \tag{41}$$

also at most $\lfloor T^{\frac{2}{3}} \rfloor$ terms in the sum of $\mathcal{E}_T(\Delta) - \mathcal{D}_T(\Delta)$ are not zero, and for any term in the exploration round $\left( \sup_{U_0/1.1 \leq \|\Delta\|_2 \leq U_0, \hat{\Theta}_{t-1}^\top \neq \boldsymbol{0} \text{ for } 1 \leq t \leq T} \left( \frac{\boldsymbol{b}_t^\top \hat{\Theta}_{t-1}^\top}{\|\boldsymbol{b}_t^\top \hat{\Theta}_{t-1}^\top\|} \Delta \boldsymbol{x}_{t,l} \right)(\boldsymbol{\delta}_t^\top \Delta \boldsymbol{x}_{t,l}) \right) -$ $\left( \inf_{U_0/1.1 \leq \|\Delta\|_2 \leq U_0, \hat{\Theta}_{t-1}^\top \neq \boldsymbol{0} \text{ for } 1 \leq t \leq T} \left( \frac{\boldsymbol{b}_t^\top \hat{\Theta}_{t-1}^\top}{\|\boldsymbol{b}_t^\top \hat{\Theta}_{t-1}^\top\|} \Delta \boldsymbol{x}_{t,l} \right)(\boldsymbol{\delta}_t^\top \Delta \boldsymbol{x}_{t,l}) \right) \leq 2U_1 \sqrt{U_2} h U_0^2$

Therefore, through Functional Hoeffding theorem (Theorem 3.26 in Wainwright (2019))), we have

$$P(\mathcal{E}_T(\Delta) - \mathcal{D}_T(\Delta) \leq -\gamma_1 | O) \leq \exp\left( -\frac{\frac{T^2}{\lfloor T^{\frac{2}{3}} \rfloor} \gamma_1^2}{16 U_1^2 U_2 h^2 U_0^4} \right) \tag{42}$$

for $\gamma_1 > 0$.

Similarly, for the exploration rounds in $\mathcal{D}_{2,T}(\Delta)$, we have

$$\left( \sup_{U_0/1.1 \leq \|\Delta\| \leq U_0} (\boldsymbol{\delta}_t^\top \Delta \boldsymbol{x}_{t,l})^2 \right) - \left( \inf_{U_0/1.1 \leq \|\Delta\| \leq U_0} (\boldsymbol{\delta}_t^\top \Delta \boldsymbol{x}_{t,l})^2 \right) \leq U_1 U_2 U_0^2 h^2. \tag{43}$$

Again, according to Functional Hoeffding theorem, we have

$$P(\mathcal{D}_{2,T}(\Delta) - \mathbb{E}(\mathcal{D}_{2,T}(\Delta)) \leq -\gamma_2 | O) \leq \exp\left( -\frac{\frac{T^2}{\lfloor T^{\frac{2}{3}} \rfloor} \gamma_2^2}{4 U_1^2 U_2^2 U_0^4 h^4} \right) \tag{44}$$

Take $\gamma_1 = \gamma_2 = 7 T^{-\frac{2}{3}} (h + h^2) (2d_x + 2d_a + 6 \log T + 6 \log L)^2 \|\Delta\|_2^2 \log T$.

Therefore,

$$P\left( \mathcal{E}_T(\Delta) - \mathbb{E}(\mathcal{D}_{2,T}) \leq -14 T^{-\frac{2}{3}} (h + h^2) (2d_x + 2d_a + 6 \log T + 6 \log L)^2 \|\Delta\|_2^2 \log T \right)$$

$$\leq P\left( \mathcal{E}_T(\Delta) - \mathcal{D}_T(\Delta) \leq -7 T^{-\frac{2}{3}} (h + h^2) (2d_x + 2d_a + 6 \log T + 6 \log L)^2 \|\Delta\|_2^2 \log T | O \right)$$

$$+ P(\mathcal{D}_{1,T}(\Delta) \leq 0 | O)$$

$$+ P\left( \mathcal{D}_{2,T} - \mathbb{E}(\mathcal{D}_{2,T}) \leq -7 T^{-\frac{2}{3}} (h + h^2) (2d_x + 2d_a + 6 \log T + 6 \log L)^2 \|\Delta\|_2^2 \log T | O \right)$$

$$+ P(O^c)$$

$$\leq \frac{1}{LT} + \frac{2}{T} + \frac{1}{T^2} \tag{45}$$

Hence with probability at least $1 - \frac{1}{LT} - \frac{2}{T} - \frac{1}{T^2}$,

$$\mathcal{E}_T(\Delta) \geq \frac{\lfloor T^{\frac{2}{3}} \rfloor}{2T} h^2 \|\Delta\|_F^2 - 14 T^{-\frac{2}{3}} (h + h^2) (2d_x + 2d_a + 6 \log T + 6 \log L)^2 \|\Delta\|_2^2 \log T. \tag{46}$$

## APPENDIX B   DETAILS ON THE SIMULATION STUDY

In this section, we detail the tuning parameters of each algorithm we used for the simulation study.

**Hi-CCAB.** There are three tuning parameters for Hi-CCAB: we set the steps for initialization $t_1 = 100$, the initialized penalization parameter $\lambda_0 = \|\frac{1}{2t_1 L} \sum_{i=1}^{t_1} \sum_{j=1}^{L} |\boldsymbol{a}_i^\top \widehat{\boldsymbol{\Theta}}_{t_1} \boldsymbol{x}_{i,j} - r_{i,j}| \boldsymbol{x}_{i,j} \boldsymbol{a}_i^\top \|_2$, and the exploration parameter $h = .1$.

**LinUCB (Li et al., 2010).** We apply the LinUCB with disjoint linear models and set multiplier for the upper confidence bound $\alpha = 1 + \sqrt{\ln(2/\delta)/2}$ with $\delta = .05$ as suggested in the paper.

**Lasso Bandit (Bastani & Bayati, 2020).** There are a couple of tuning parameters in the original algorithm including $h$ for the set of "near-optimal arms", $q$ for the force-sample set, and $\lambda_1$ and $\lambda_{2,0}$ as the regularization parameters for the "forced sample estimate" and "all-sample estimate". We follow the original paper and set $h = 5$, $\lambda_1 = \lambda_{2,0} = 0.05$. We set $q = 2$ so that the size of initialized forced sample set is close to that we used for Hi-CCAB.

**NeuralUCB (Zhou et al., 2020).** The tuning parameters of NeuralUCB include the confidence parameter as in all UCB-based algorithm, the size of neural network, as well as the step size, regularization parameter for gradient descent to train the neural network. We adapted the code from https://github.com/uclaml/NeuralUCB and used the default settings.

**EE-Net (Ban et al., 2022).** EE-Net involves tuning parameters for gradient descent to train the exploitation network, exploration network, and the decision-maker network. We adapted the code from https://github.com/banyikun/EE-Net-ICLR-2022 and used the default settings.

## APPENDIX C  MORE DETAILS ON THE CASE STUDY AND ADDITIONAL NUMERICAL RESULTS

In this section, we provides more background information on the case study and additional interpretations of the represetation matrix $\boldsymbol{\Theta}$ and numerical results.

Figure 3a shows the daily sales by product and each color represents one product (only products that appeared more than 95% of the days are colored; the rest are colored as grey). The days corresponding to the vertical dashed grey lines are days with promotion. The two red vertical lines correspond to the annual sales events. The variation between products was large and one product dominated the rest most of the time. The sales were also driven by the promotion – the sales went up when there is a promotion. Figure 3b shows the median unit price across time with the 25th and 75th quantiles as the boundaries of the grey area. The median unit price was around 3.2 RMB and there were variations in unit price among products. Figure 3c shows the number of single-flavor and multi-flavor products. Three-quarters of the products were single-flavored. Note that products with the same flavor can have different package sizes. Figure 3d shows the number of products with different package sizes. The package size of about 60% of the products is larger than 20 with 30% having package sizes between 10 and 20 and the rest less than 10.

Figure 4 compares the real sales with the simulated sales based on model (1) using the pseudo ground truth $\boldsymbol{\Theta}$ and $\sigma$ that we estimated using all the data, i.e., the teal line is $\sum_{j=1}^{L} r_{t,j}$ where $r_{t,j}$ is generated by

$$r_{t,j} = \boldsymbol{a}_t^\top \boldsymbol{\Theta} \boldsymbol{x}_{t,j} + \varepsilon_{t,j}, \quad \varepsilon_{t,j} \sim N(0, \sigma^2)$$

where $\boldsymbol{a}_t$ and $\boldsymbol{x}_{t,j}$ are from the real data. As shown in Figure 4, the real sales and the simulated sales follow quite closely across time, which indicates that both our model and estimation are reasonable.

**Structure of the representation matrix $\boldsymbol{\Theta}$.** One advantage of model is the interpretability which allows us to gain insights from the representation matrix $\boldsymbol{\Theta}$. Specifically, our model is able to discover the underlying factors of the effect of both arms and covariates on the reward. In the following, we will examine the pseudo ground truth $\boldsymbol{\Theta}$ we obtained using all the data.

The rank of $\boldsymbol{\Theta}$ is 5 with the singular values being $(2.5, 0.3, 0.2, 0.02, 0.002)$. The leading singular value dominates the rest and thus the leading left and right singular vectors are the most important ones in explaining the effect on the reward and we focus on the leading singular vectors in what follows.

Figure 5 shows the loadings for different covariates (i.e., the leading right singular vector) and our algorithm is able to learn interpretable patterns of the effects on the reward – for weekday, the effects are drastically different during the weekend and during the weekend; for months, the effects show different patterns during the promotion month (June and November) from other months; for location, the effects of the coastal provinces are different from the rest, which exactly corresponds to the levels of economic development of different regions in China. In sum, our model can exploit the underlying structure of the covariates and provide insights into purchasing behavior and seasonality.

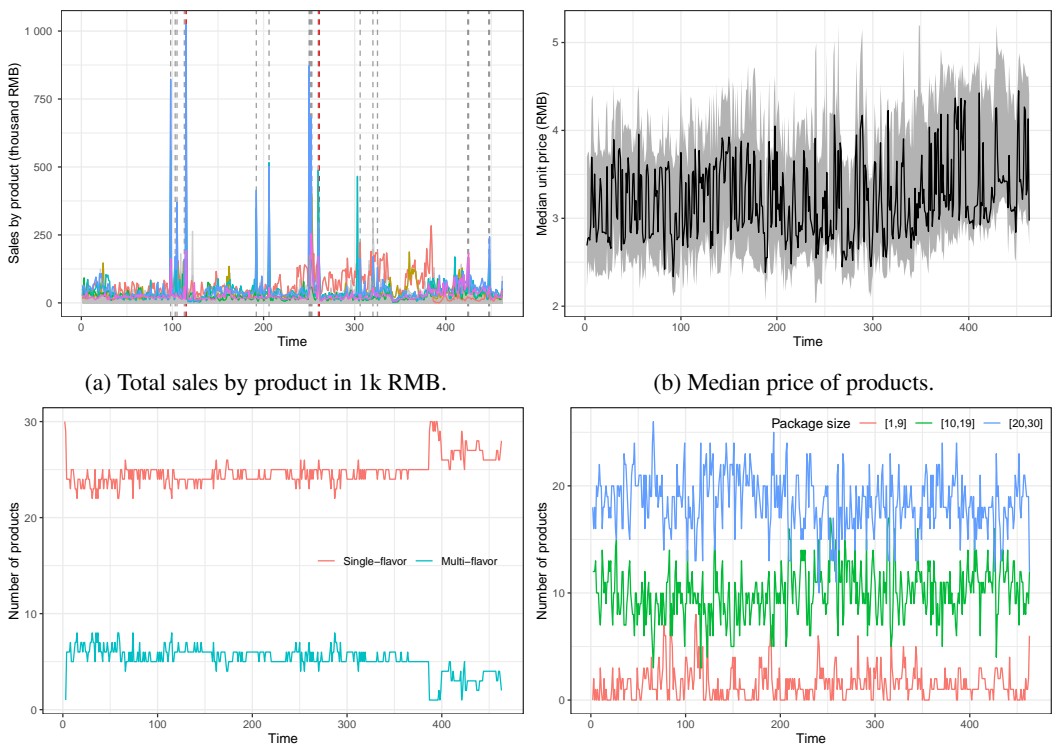

(a) Total sales by product in 1k RMB.

(b) Median price of products.

(c) Number of products with various number of flavors. (d) Number of products with various package sizes.

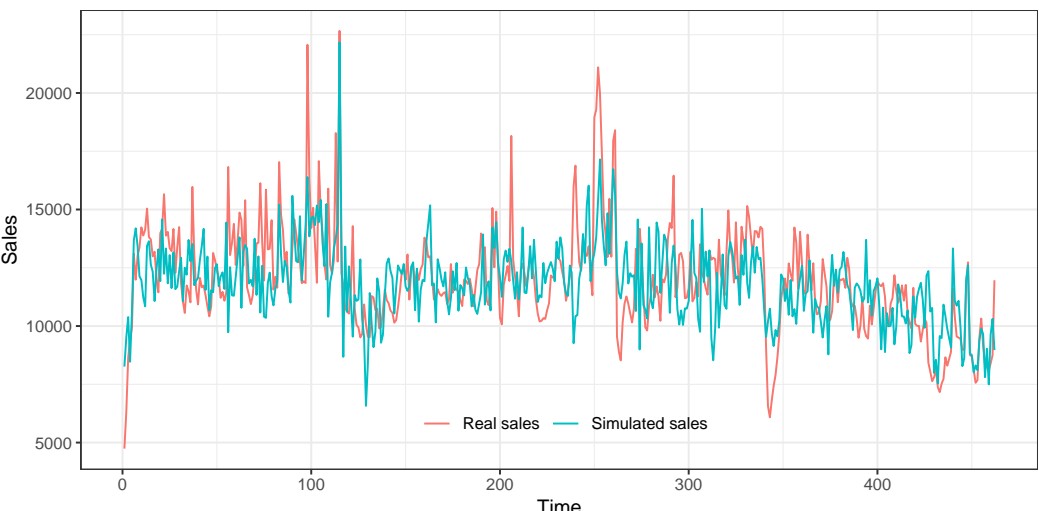

Figure 4: Real sales vs simulated sales.

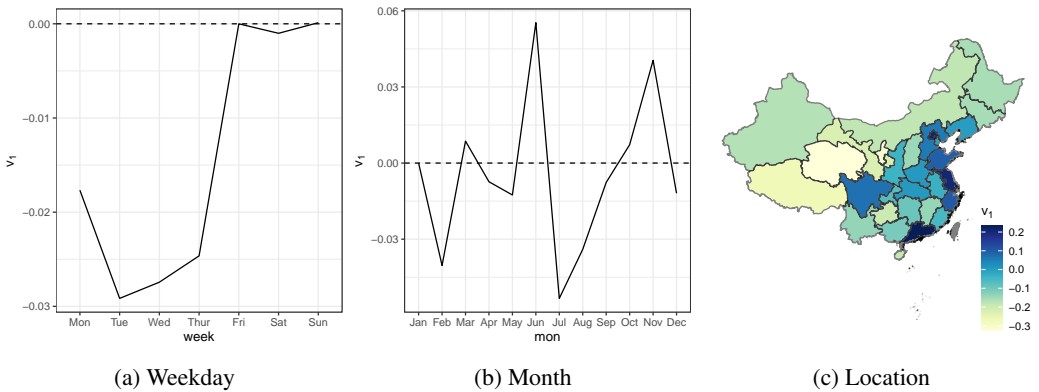

(a) Weekday           (b) Month           (c) Location

Figure 5: Loadings of the leading right singular vectors for the covariates.

On the other hand, Table 1 explores the loadings for the arm on May 29th 2022, the last Sunday in our data (i.e., the leading left singular vectors multiplied with $\langle \boldsymbol{v}_1, \bar{\boldsymbol{x}} \rangle$ where $\bar{\boldsymbol{x}}$ is the average of $\boldsymbol{x}_j$ for $j = 1, \ldots, L$ on May 29th 2022). Specifically, we investigate the effect of flavors on the reward given the context. We take the average of the loadings of the linear and quadratic terms for each flavor in all 30 products and compare with the total sales of each flavor across all Sundays in Mays. For ease of comparison, we further scale the sales and the loadings by their corresponding largest numbers. The loadings and sales are closely related to each other.[1] As in Table 1, on May 29th 2022, flavor 1 (F1) has the largest effect, followed by flavor 10, 13, 7, 9 and 11. Therefore, our model learns the values of the flavors (per unit).

|  | F1 | F2 | F3 | F4 | F5 | F6 | F7 | F8 | F9 | F10 | F11 | F12 | F13 |
|---|---|---|---|---|---|---|---|---|---|---|---|---|---|
| Sales | 1.00 | 0.05 | 0.00 | 0.00 | 0.00 | 0.03 | 0.19 | 0.00 | 0.08 | 0.19 | 0.18 | 0.00 | 0.38 |
| $\tilde{u}_1$ (linear) | 1.00 | 0.00 | 0.00 | 0.00 | 0.00 | 0.00 | 0.00 | 0.00 | 0.00 | 0.00 | 0.00 | 0.00 | 0.00 |
| $\tilde{u}_1$ (quadratic) | 1.00 | 0.12 | -0.00 | 0.00 | 0.00 | 0.03 | 0.19 | 0.00 | 0.15 | 0.39 | 0.16 | 0.03 | 0.33 |

Table 1: Total sales and loadings of the linear and quadratic terms (scaled) of the 13 flavors.

**More on simulation with additional numerical results.** We first detail how we ran the simulation and then provide more simulation results. To be specific, we first use $t_1 = 100$ for the initialization step to estimate $\widehat{\boldsymbol{\Theta}}_{t_1}$; and then at each time $t = t_1 + 1, \ldots, T$, we follow Algorithm 1 to decide on the action $\boldsymbol{a}_t$ for assortment and pricing. After determining $\boldsymbol{a}_t$, we generate the sales $\boldsymbol{r}_t$ according to (1) using the pseudo true $\boldsymbol{\Theta}$ and $\sigma$. We further compare the performance of the assortment-pricing policy with exploration and without exploration and with different initialization time $t_1$. Each setup is simulated 100 times.

Figures 6a-6b show cumulative regret and Figures 6d show percentage gain in cumulative sales when $t_1 = 20, 50, 100$ with exploration and without exploration. Hi-CCAB with exploration performs better then without exploration. As expected, longer initialization steps provide a better initial estimation of the $\boldsymbol{\Theta}$ and thus helps with the performance in a short time windows. As time goes by, the time-averaged cumulative regret all converge to zero and the percentage gain in cumulative sales should converge.

---

[1]The correlation of sales and the linear-term loadings is 0.91 and that of the quadratic-term loadings is 0.97.

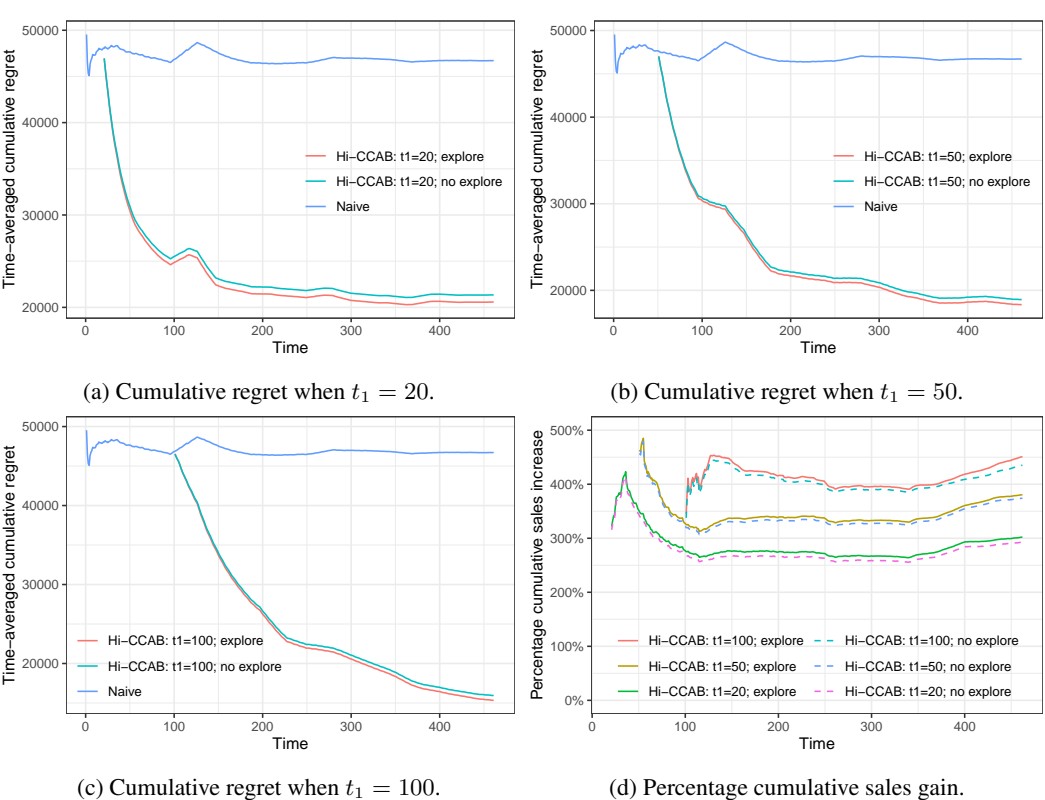

(a) Cumulative regret when $t_1 = 20$.

(b) Cumulative regret when $t_1 = 50$.

(c) Cumulative regret when $t_1 = 100$.

(d) Percentage cumulative sales gain.

Figure 6: Performance of `Hi-CCAB` with different initialization times $t_1$ and with exploration and without exploration.

