# OpenReview forum: "High-dimensional Continuum Armed and High-dimensional Contextual Bandit: with Applications to Assortment and Pricing"
_ICLR.cc/2023/Conference — Submitted to ICLR 2023_

### Official Review · Reviewer_K6HK · 2022-10-24

**Confidence:** 3
**Clarity, Quality, Novelty And Reproducibility:** 1. The paper is well written but the …
**Correctness:** 2
**Technical Novelty And Significance:** 3
**Empirical Novelty And Significance:** 2
**Recommendation:** 5

**Strength And Weaknesses:**

 However, I have the following concerns about the exploration manner, related works and experimental design.
(1)	Although the proposed model is based the low rank representation matrix assumption, I cannot get the reason why the epsilon term is bounded. Previous works usually assume a sub-gaussian distribution for the error term. I suggest the authors to explain why the low-rank assumption is enough to bound the epsilon.

(2)	The paper gives two different manner (2 different gaussian distribution) for exploration in each step. Theorem 1 shows the regret bound based on the first one. What about the another one? Is it possible for us to propose a general regret bound which is independent on the distribution used in the exploration?

(3)	There are some important related works are missing. Recently the direction of proposing deep learning based model for high dimensional contextual bandit problem is very hot, especially for the complex non-linear model assumption. For example,
[1] Zhou D, Li L, Gu Q. Neural contextual bandits with ucb-based exploration[C]//International Conference on Machine Learning. PMLR, 2020: 11492-11502.
[2] Chen Y, Xie M, Liu J, et al. Interconnected Neural Linear Contextual Bandits with UCB Exploration[C]//Pacific-Asia Conference on Knowledge Discovery and Data Mining. Springer, Cham, 2022: 169-181.
[3] Jacot, A., Gabriel, F., Hongler, C.: Neural tangent kernel: convergence and generalization in neural networks. In: NIPS, pp. 8580–8589 (2018)
If we concatenated the continuum arm space and contextual vector totally as a large feature vector, we can use these existing methods to solve it. Thus the paper should be compared to them.

(4)	The experiment study is a toy. Only one navie method is compared to Hi-CCB with the simulation in a small dataset. Thus it is very hard to support the conclusion. I suggest the authors to justify by some public large data for real problem. In fact, we can compare with baselines in each of special case problems.

Minor problem:
(1)	All figures are blur.


**Summary Of The Paper:**

This paper proposes a general framework to solve the high dimensional continuum armed bandit problem and high dimensional contextual bandit problem jointly. This research problem is significant and the proposed model is novel. I agree that the proposed problem has broad scope of applications. For this problem, the paper propose an efficient algorithm called Hi-CCAB, by adopting the low-rank matrix estimator. Experimental study shows the proposed method can improve the naive approach by 4 times for the cumulative reward.

**Summary Of The Review:**

1. The problem is significant and interesting but the assumption remained to be listed clear.
2. The experimental study should be improved. The current version is not convincing.
3. The exploration manner should be explained and regret bound could be proved accordingly.

---

> ### Author Response · Authors · 2022-11-19
> **Response to Reviewer K6HK (1/2)**
>
>
> Thank you for your positive and constructive comments.
> We spent major efforts in revising the theoretical portions and providing more simulation study and we hope the revised version is more convincing.
>
>
> > **Q**: Why the epsilon term is bounded? Why the low-rank assumption is enough to bound the epsilon?
>
> **A**:
> Assume you are referring to the error term in Lemma 1, the bound of it does not depend on the low-rank property. Lemma 1 establishes a restricted-strong-convexity type result. The place that the low-rank property comes in is in the main proof where we use Lemma 1, Lemma 2, and low-rank property to give tail bound on estimation error of $\boldsymbol{\Theta}$, namely, $\delta \boldsymbol{\Theta}$. We show the proof sketch in the following to provide more insights.
>
> There are two major steps in the proof: (1) bounding the estimation error for the low-rank representation matrix estimator; (2) bounding the expected cumulative regret.
>
> (1) Bounding the estimation error of $\widehat{\boldsymbol{\Theta}}_t$ with a high probability bound. Denote $\delta\boldsymbol{\Theta}_t = \widehat{\boldsymbol{\Theta}}_t - \boldsymbol{\Theta}$.
>  We show that for a large $t$,
>
>  $$P\left(||\delta \boldsymbol{\Theta}_t||_F \le {3\over T^{2\over 15}} {\sigma\sqrt{d_x+1} \over \sqrt{L}h^2} + 6\lambda_0{\sqrt{2d}\over h^2T^{1\over 6}}\right) \le 1 - \left( {3\over t} + {2\over t^2} + {2\over Lt} + {2\over L^3t^3} + {1\over t^{2\over 15}}  \right).$$
>
> Note that the action taken is based on previous estimators and affects the accuracy of future estimators, leading to lots of dependencies. The classical matrix completion results can no longer apply. Through careful use of conditional expectations, martingales, and empirical process we separate out different sources of randomness from the exploration and the covariates (i.e., $\delta_1,\cdots, \delta_t, x_{1,\cdot},\cdots,x_{t,\cdot}$) to derive the bounds. Lemma 1 establishes a restricted-strong-convexity-type result of the sum of squares in the objective function. Lemma 2 establishes a Lipschitz-type result of the sum of squares in the objective function.  Further analysis of the nuclear-norm-penalized sum of squares with the two lemmas and low-rank properties gives the tail bound of the estimation error.
>
> (2)
> Bounding the time-averaged expected regret. Let $Q_t = \{||\delta \boldsymbol{\Theta}_t||_F \le {3\over T^{2\over 15}} {\sigma\sqrt{d_x+1} \over \sqrt{L}h^2} + 6\lambda_0{\sqrt{2d}\over h^2T^{1\over 6}}  \}$ be the event such that $\delta \boldsymbol{\Theta}_t$ is bounded. We know that from the first step, for large $t$, $P(Q_t^c) \le {3\over t} + {2\over t^2} + {2\over Lt} + {2\over L^3t^3} + {1\over t^{2\over 15}} $. Consider the expectation of the regret on $Q_t$ and $Q_t^c$ separately and both terms vanish with $t$ at the polynomial rate.
>
>
>
> ----
>
> > **Q**: A general regret bound which is independent on the distribution used in the exploration?
>
> **A**:
> Thanks for this very intriguing question. The analysis of regret with the first exploration method is already very hard as it involves intertwined dependency. The second way for exploration introduces additional dependency (as it depends on the previous actions, which depends on estimations prior to that, which further depends on actions even prior to that, and so on and so forth). So it is even harder to rigorously analyze. Regardless of this, a rough sense/intuition of its performance is available. As the second way of exploration is primarily to avoid tuning parameter $h$ while having the right scale, its performance is expected to be similar to that with $h$ taking the value of the true standard deviation of the optimal arm (rather than the empirical version) if the different axis of the arm is homogeneous.
>
>
>
> ----

---

> ### Author Response · Authors · 2022-11-19
> **Response to Reviewer K6HK (2/2)**
>
>
>
> > **Q**: Related work on deep learning based model for high dimensional contextual bandit problem.
>
> **A**: Thank you for pointing out the literature in the deep learning based contextual bandit algorithms.
>
> To our knowledge, neural bandits are not for continuous arm  space but rather for discrete arm space. Since discrete space to continuous space does not have onto mapping, we are not sure how the concatenating works to provide a working method for our problem setting.
>
> Also, our method is designed to work in business and healthcare settings, where the data and the opportunity for intervention are not unlimited and models that can draw further interpretations are preferable.
>
> Regardless of this, while our method is designed for the high-dimensional continuum armed and high-dimensional contextual bandit, it is also compatible with the regular multi-armed contextual bandit. So we compare our methods with two deep-learning based contextual bandit algorithms NeuralUCB (Zhou et al., 2020) and EE-Net (Ban et al., 2022) in Section 4 under the multi-armed bandit setup. Our method outperforms those two deep learning bandit methods.
>
>
> ----
>
>
> > **Q**: More experiment study.
>
> **A**:
> We compare Hi-CCAB with  LinUCB (Liet al., 2010), Lasso Bandit (Bastani \& Bayati, 2020),  NeuralUCB (Zhou et al., 2020), and EE-Net (Ban et al., 2022) in Section 4 under the multi-armed bandit setup and we consider a non-sparse case and a sparse case.
> As shown in our simulation results, Hi-CCAB outperforms the other three algorithms in the non-sparse case
> and performs well in the sparse case, which is not to the advantage of Hi-CCAB.
> It only falls short of the Lasso bandit that is specifically designed for the sparse setting when the number of arms is relatively small. It still outperforms all the others when the number of arms becomes large.
>
>
> ----
>
>
> > **Q**: Blurry figures.
>
> **A**: We apologize for the inconvenience but we did not find the same problem from the pdf downloaded from the web. We hope the revised will not have this problem.

---

### Official Review · Reviewer_q4F5 · 2022-10-24

**Confidence:** 3
**Correctness:** 2
**Technical Novelty And Significance:** 1
**Empirical Novelty And Significance:** 2
**Recommendation:** 3

**Clarity, Quality, Novelty And Reproducibility:**

The novelty and contribution of this paper are limited, especially the theoretical guarantee, and the writing of the paper can be improved.

**Strength And Weaknesses:**

Strengths:
The paper studies an interesting and relevant problem and proposes a model that may be applied to a broader class of problems.

Weaknesses:
1. The novelty and contribution of this paper are limited, especially the theoretical guarantee.
2. The writing of the paper can be improved.

Detailed comments:
1. The first sentences in the abstract are essentially saying "we formulate a bandit problem as a bandit problem".
2. Line 5 of the abstract: "contextual" -> "context"
3. On page 3, is the concept "approximately low rank" defined?
4. In (1), are the contexts stochastic (they appear to be so from the definition of the regret)?
5. On page 5, it is not justified why in the healthcare example one would expect a low-rank matrix.
6. Theorem 1:
 (1) the result is not very informative in that one can not see the dependence on $d_x$, $d_a$ and $d$ (i.e., how using low-rank representation learning can address the issue of high dimensionality).
 (2) As T goes to infinity, are $d_x$ and $d_a$ also going to infinity? Since the difficulty of high-dimensionality mainly
stems from the fact that $d\ge T$. It seems to me from the proof that at least $d_x$ is treated as fixed as $T \rightarrow \infty$. Please correct me if I am wrong.



**Summary Of The Paper:**

The paper considers the problem of a contextual bandit with both a high-dimensional action space and a high-dimensional covariate space. The author(s) propose a new bandit model describing the aforementioned problem, and provide a learning algorithm under the model.  Theoretical results are developed for the algorithm and the effectiveness of the algorithm is demonstrated with an assortment pricing case study.

**Summary Of The Review:**

In general, I think the paper considers an interesting problem and proposes an appropriate model describing the problem. The paper however can be improved with improved theoretical results and presentation.

---

> ### Author Response · Authors · 2022-11-19
> **Response to Reviewer q4F5 (1/2)**
>
>
> Thank you for your constructive comments.
> We have spent major efforts to improve the theoretical results and revamp the presentation based on your comments and questions. We copy the updated version of our main theorem here which shows explicit dependence on the dimensions and rank, and will address your specific questions later.
>
> **Theorem 1:** Suppose $x_{t,l} \overset{i.i.d}{\sim}  N(0_{d_x} , I_{d_x})$, and the errors ($\varepsilon_{t,j}$) defined in Model (1) follow normal distributions: $\varepsilon_{t,j} \overset{i.i.d}{\sim} N (0,\sigma^2) $. Suppose $\boldsymbol{\Theta}$ is rank $d$. Suppose the exploration step in Algorithm 1 is $a_t  = \hat{a}_t + \delta_t$ for $t \in$ {$\{ \lfloor w^{3\over 2}\rfloor : w\in \mathbb{Z}+  \}$} where $\delta_t \sim N(0, h I) $, $\mathcal{A}_t =${$\{a \in R^{d_a}:||a||\le 1\} $}, then there is a $T_1$ such that for $T\ge T_1$, the expected cumulative regret of the Algorithm 1, $R_T^{\pi}$, satisfies
>
> \begin{equation}
>     {R_T^{\pi} \over T}   \le {2\sqrt{Ld_x} ||\boldsymbol{\Theta}||_2  T_1  } T^{-1} +  {72\over 5}\lambda_0{\sqrt{ 2dd_xL  }\over h^2 }T^{-{1\over 6}} + {60\over 13 }\sqrt{Ld_x}||\boldsymbol{\Theta}||_2 T^{-{2\over 15}}+ {90 \over 13}{\sigma(d_x+1) \over h^2}T^{-{2\over 15}} ,
> \end{equation}
>
> where $T_1 = C_{h,L,\lambda_0} (d_x+d_a)^6 \left(log(d_x+d_a)\right)^3 $ and the constant $C_{h,L,\lambda_0}$ depends on $h$, $L$ and $\lambda_0$.
> For $T\le T_1$, ${R_T^{\pi} \over T}   \le 2\sqrt{Ld_x}||\boldsymbol{\Theta}||_2$.
>
> We will answer your specific questions below.
>
> > **Q**: Is the concept "approximately low rank" defined?
>
> **A**:
> Thanks for pointing out our typo, we mean low-rank.
>
> ----
>
>
> > **Q**: Are the contexts stochastic?
>
> **A**: Yes we assume the contexts are stochastic in Theorem 1 for the convergence rate of the regret. However, our algorithm does not require stochastic contexts.
>
> ----
>
> > **Q**: Why in the healthcare example one would expect a low-rank matrix?
>
> **A**: In health care/electronic health records, lots of measurements are taken to characterize one person, while only a few really matter. The possible actions are also related (e.g., having similar health effects), and only a few out of many interventions are really useful. This is where the low-rank structure comes in.
>
>
> ----
>
>
> > **Q**: Dependence on $d_x$, $d_a$ and $d$?
>
> **A**: The updated theorem now shows the explicit dependence on the dimensions $d_x$, $d_a$, and rank $d$.
> When $T$ is small, the ``burnout'' term (the first term) dominates.
> It depends on $T$ and the dimensions but not the rank as ${(d_a+d_x)^6 (log(d_a+d_x))^3 T^{-1}}$, whose order depends on the exponent defining the exploration set (i.e., how frequent we explore). As $T$ grows, the second term dominates. $\lambda_0$ is of order $\sigma\sqrt{d_x}$, so the second terms depends on $T, d_x$ and $d$ but not $d_a$ at the order of $\Omega({d_x\sqrt{d} T^{-{1\over 6}}})$.
> Without the low-rank assumption, the order would be
> $\Omega({d_x^{3\over 2} T^{-{1\over 6}}})$ instead.
> When $T$ becomes even larger, the last two terms dominate, at the order $\Omega( {d_x T^{-{2\over 15}}} )$. However, the last case rarely happens, as it requires the order of $T$ equal to or larger than $d^{15}$.
> Therefore, taking dimensions and rank into consideration, the time-averaged expected cumulative regret is mostly at the order of $\Omega({d_x\sqrt{d} T^{-{1\over 6}}})$.
>
> ----

---

> ### Author Response · Authors · 2022-11-19
> **Response to Reviewer q4F5 (2/2)**
>
>
>
> > **Q**: How using low-rank representation learning can address the issue of high dimensionality?
>
> **A**:
> The low-rank representation helps in the intermediate estimation, in particular, in providing the tail bound of the estimation error for representation matrix $\boldsymbol{\Theta}$ together with Lemma 1 and Lemma 2.
> At the same time, as in the response to the previous question, the order would be
> $\Omega({d_x^{3\over 2} T^{-{1\over 6}}})$ instead of $\Omega({d_x\sqrt{d} T^{-{1\over 6}}})$ without the low-rank assumption.
> We
>
> show the proof sketch in the following to provide more insights.
>
> There are two major steps in the proof: (1) bounding the estimation error for the low-rank representation matrix estimator; (2) bounding the expected cumulative regret.
>
> (1) Bounding the estimation error of $\widehat{\boldsymbol{\Theta}}_t$ with a high probability bound. Denote $\delta\boldsymbol{\Theta}_t = \widehat{\boldsymbol{\Theta}}_t - \boldsymbol{\Theta}$.
>     We show that for a large $t$,
>
> $$P\left(||\delta \boldsymbol{\Theta}_t||_F \le {3\over T^{2\over 15}} {\sigma\sqrt{d_x+1} \over \sqrt{L}h^2} + 6\lambda_0{\sqrt{2d}\over h^2T^{1\over 6}}\right) \le 1 - \left( {3\over t} + {2\over t^2} + {2\over Lt} + {2\over L^3t^3} + {1\over t^{2\over 15}}  \right).$$
>
> Note that the action taken is based on previous estimators and affects the accuracy of future estimators, leading to lots of dependencies. The classical matrix completion results can no longer apply. Through careful use of conditional expectations, martingales, and empirical process we separate out different sources of randomness from the exploration and the covariates (i.e., $\delta_1,\cdots, \delta_t, x_{1,\cdot},\cdots,x_{t,\cdot}$) to derive the bounds. Lemma 1 establishes a restricted-strong-convexity-type result of the sum of squares in the objective function. Lemma 2 establishes a Lipschitz-type result of the sum of squares in the objective function.  Further analysis of the nuclear-norm-penalized sum of squares with the two lemmas and low-rank properties gives the tail bound of the estimation error.
>
>
> (2)
> Bounding the time-averaged expected regret. Let $Q_t = \{||\delta \boldsymbol{\Theta}_t||_F \le {3\over T^{2\over 15}} {\sigma\sqrt{d_x+1} \over \sqrt{L}h^2} + 6\lambda_0{\sqrt{2d}\over h^2T^{1\over 6}}  \}$ be the event such that $\delta \boldsymbol{\Theta}_t$ is bounded. We know that from the first step, for large $t$, $P(Q_t^c) \le {3\over t} + {2\over t^2} + {2\over Lt} + {2\over L^3t^3} + {1\over t^{2\over 15}} $. Consider the expectation of the regret on $Q_t$ and $Q_t^c$ separately and both terms vanish with $t$ at the polynomial rate.
>
>
>
> ----
>
>
> > **Q**: As $T$ goes to infinity, are $d_x$ and $d_a$ also going to infinity and what are the relationship?
>
> **A**:
> All $T$, $d_x$, $d_a$ and $d$ can grow to infinity. Unlike the pure estimation problem where the criterion is on estimation accuracy, our problem seeks to give good actions such that the time-averaged expected cumulative regret is minimized. So this is a more difficult problem in that 1, good estimation only matters through its influence on actions taken later, and 2, we can not have actions perfect for facilitating estimation at risk of incurring larger regret. So the order of dimension is no longer larger than $T$ but rather of a similar or smaller order (i.e., $T^{1\over 6}$ for our current algorithm configuration, but can be changed to a larger one with the change of exponent number defining exploration set).
>
>
>
> ----

---

### Official Review · Reviewer_yX2p · 2022-10-24

**Confidence:** 3
**Correctness:** 3
**Technical Novelty And Significance:** 2
**Empirical Novelty And Significance:** 3
**Recommendation:** 5

**Clarity, Quality, Novelty And Reproducibility:**

Clarity and quality:
The main text is well written, though I didn't check all the detail in the appendix.

Novelty:
The main novelty comes from the policy learning phase in my opinion, which is enforced by a low-rank matrix learning process.

Reproducibility:
If I'm not missing something, the reproducibility is not highlighted enough in terms of data collection. But it should be ok if the data is made available later. Otherwise, the experiments detail is given.

**Strength And Weaknesses:**

Strength:
- The proposed model can be reduced to some classic bandit problems (such as stochastic multi-armed bandit, linear bandit, continuum bandit without context, etc).
- Very original dataset that corresponds to the purpose of the paper.
- The policy learning phase which expands the action space is quite interesting to deal with the continuum arm space. And the two phases (low-rank representation learning + policy learning) although interleaved, seem to be quite modular.

Weaknesses/questions/remarks:
- Could you elaborate on the choice of $w^{3/2}$?
- The theoretical part is not very strong in the sense that we do not really know how to assess Theorem 1. I would suggest the authors to interpret a bit Theorem 1 since it is not very common to look at the 'time-averaged regret', so some interpretation would be appreciated. In particular, could elaborate on the order $1/13$, and how is it related to the dimensions?
- On the experiment side, as I mentioned I find the proposed two-step algorithm quite modular. That's a good point, but at the same it means that we can also try other kind of contextual bandit algorithms (or even kernel/graph/combinatorial bandit algorithms), at least as some benchmark to compare with.
- Another point on the experiment - as the authors claim that the proposed model covers also some other more classic bandit models, a small sanity check (both experimentally and theoretically actually) that could've been done is to check the performance of the proposed algorithm under these classic models.
- Literature review is somehow not very thorough (at least on the bandit side, contextual bandits, continuum bandits, rank-1 bandits etc), but that's a minor point.

**Summary Of The Paper:**

This paper proposes a low-rank representation matrix to address the high-dimensional continuum-armed and high-dimensional contextual bandit problem motivated by assortment pricing problems. The idea is to use an interleaved two-phases algorithm where the first step is to learn a low-rank representation matrix to capture the low-dimensional structure of both the high-dimensional arms and high-dimensional covariates, and a second step to expand the action space. On the theory side, a "consistency" result is given ($\lim_{T->\infty} \frac{R_T}{T^{12/13}} = 0$), and on the practical side, experiments on and instant-noodle assortment pricing dataset are provided.

**Summary Of The Review:**

The idea is in general interesting, in particular the policy learning phase. However, I think there could be a set of things to be done to make the paper more complete as I already mentioned in the Strength And Weaknesses section.

___________________________________________

I decide to increase a bit my score after rebuttal since the authors have made an effort to address some of the points made by other reviewers and myself. Those are, however, some major revisions (that are not supposed to be made during the rebuttal phase, otherwise that's like extending the deadline).

---

> ### Author Response · Authors · 2022-11-19
> **Response to Reviewer yX2p (1/2)**
>
>
> Thank you for your constructive comments and point out the originality of our dataset.
> Indeed, it is real sales data that motivates us to work on this important problem that received little attention in the literature.
> We report the changes we made in the revision to address your comments and make both the theoretical and the experiment sections more convincing.
>
>
> > **Q**: Choice of $w^{3/2}$?
>
> **A**: The intuition for $w^{3/2}$ is to explore more in the initial stage and exploit more in the later stage of the algorithm. Because in the initial steps, gaining insights into the representation matrix is more important than choosing the best arm on the ground of the scarce knowledge of the situation.
> A similar strategy is also adopted in other literature such as Bastani \& Bayati (2020) for the Lasso Bandit.
>
> The exponent ${3\over 2}$ in the exploration set {$\{\lfloor w^{3\over 2} \rfloor: w \in \mathbb{Z}_+\}$} is not essential, any number larger than 1 would work, although the convergence rate will be changed as discussed in the response to your next question. The polynomial form is not essential either. As long as it aligns with the intuition mentioned above, the time-average cumulative regret will still converge.
>
> ----
>
> > **Q**: "time-average regret" vs "regret"?
>
> **A**: We consider the time-averaged expected cumulative regret since it measures the trend of the newly incurred regret, in the long run, more directly; on the other hand, the expected cumulative regret grows with time $T$, which is less interpretable. If we multiply the time-averaged expected cumulative regret by $T$, we will get the expected cumulative regret.
>
>
> ----
>
> > **Q**: Interpretation of Theorem 1 and relationships between the dimensions and time.
>
> **A**: As other existing methods are not applicable to the problem we consider, since we facing (1) the double-layer of the high dimensionality of the arm space and the covariate space and (2) continuous arm space, it is hard to compare the convergence rates across methods.
>
> To address your concern, we update Section 3.2 theoretical results: 1) improve the rate and 2) explicitly show the dependence of the rate on the dimensions and rank.
>
> **Theorem 1:** Suppose $x_{t,l} \overset{i.i.d}{\sim}  N(0_{d_x} , I_{d_x})$, and the errors ($\varepsilon_{t,j}$) defined in Model (1) follow normal distributions: $\varepsilon_{t,j} \overset{i.i.d}{\sim} N (0,\sigma^2) $. Suppose $\boldsymbol{\Theta}$ is rank $d$. Suppose the exploration step in Algorithm 1 is $a_t  = \hat{a}_t + \delta_t$ for $t \in$ {$\{ \lfloor w^{3\over 2}\rfloor : w\in \mathbb{Z}+  \}$} where $\delta_t \sim N(0, h I) $, $\mathcal{A}_t =${$\{a \in R^{d_a}:||a||\le 1\} $}, then there is a $T_1$ such that for $T\ge T_1$, the expected cumulative regret of the Algorithm 1, $R_T^{\pi}$, satisfies
>
> \begin{equation}
>     {R_T^{\pi} \over T}   \le {2\sqrt{Ld_x} ||\boldsymbol{\Theta}||_2  T_1  } T^{-1} +  {72\over 5}\lambda_0{\sqrt{ 2dd_xL  }\over h^2 }T^{-{1\over 6}} + {60\over 13 }\sqrt{Ld_x}||\boldsymbol{\Theta}||_2 T^{-{2\over 15}}+ {90 \over 13}{\sigma(d_x+1) \over h^2}T^{-{2\over 15}} ,
> \end{equation}
>
> where $T_1 = C_{h,L,\lambda_0} (d_x+d_a)^6 \left(log(d_x+d_a)\right)^3 $ and the constant $C_{h,L,\lambda_0}$ depends on $h$, $L$ and $\lambda_0$.
> For $T\le T_1$, ${R_T^{\pi} \over T}   \le 2\sqrt{Ld_x}||\boldsymbol{\Theta}||_2$.
>
>
> **Interpretation of the convergence rate:**
> An intuitive understanding of Theorem 1 is that the expected regret incurred each time converges to zero at a speed at least $T^{-{2\over 15}}$ as $T$ going to infinity.
>  The convergence rate depends on the frequency of the exploration which depends on the exponent $3\over 2$ in the exploration set, {$\lfloor w^{3\over 2} \rfloor: w \in \mathbb{Z}_+$}.
>  Recall that the exponent can be changed with any number larger than 1, which can be considered a tuning parameter.
>
>
> **Dependence on dimensions $d_a, d_x$ and rank $d$:**
>  When $T$ is small, the ``burnout'' term (the first term) dominates.
> It depends on $T$ and the dimensions but not the rank as ${(d_a+d_x)^6 (log(d_a+d_x))^3 T^{-1}}$, whose order depends on the exponent defining the exploration set (i.e., how frequent we explore). As $T$ grows, the second term dominates. At the same time, $\lambda_0$ is of order $\sigma\sqrt{d_x}$, so the second terms depends on $T, d_x$ and $d$ but not $d_a$ at the order of $\Omega({d_x\sqrt{d} T^{-{1\over 6}}})$.
> Without the low-rank assumption, the order would be
> $\Omega({d_x^{3\over 2} T^{-{1\over 6}}})$ instead.
> When $T$ becomes even larger, the last two terms dominate, at the order $\Omega( {d_x T^{-{2\over 15}}} )$. However, the last case rarely happens, as it requires the order of $T$ equal to or larger than $d^{15}$.
> Therefore, taking dimensions and rank into consideration, the time-averaged expected cumulative regret is mostly at the order of $\Omega({d_x\sqrt{d} T^{-{1\over 6}}})$.

---

> ### Author Response · Authors · 2022-11-19
> **Response to Reviewer yX2p (2/2)**
>
> > **Q**: Comparison with classic bandit models and other contextual bandit algorithms.
>
> **A**:
> Indeed our model covers many classic bandit models as they are special cases of our model, and thus our method applies to their settings.
>
> For empirical comparison, we compare Hi-CCAB with  LinUCB (Liet al., 2010), Lasso Bandit (Bastani \& Bayati, 2020),  NeuralUCB (Zhou et al., 2020), and EE-Net (Ban et al., 2022) in Section 4 under the multi-armed bandit setup and we consider a non-sparse case and a sparse case.
> As shown in our simulation results, Hi-CCAB outperforms the other three algorithms in the non-sparse case
> and performs well in the sparse case, which is not to the advantage of Hi-CCAB.
> It only falls short of the Lasso bandit that is specifically designed for the sparse setting when the number of arms is relatively small. It still outperforms all the others when the number of arms becomes large.
>
> For theoretical comparison, each bandit model has its own specific assumptions. It would take lots of work and space to systematically compare our methods under their settings one by one, which is also diverging from our main points. At least, the empirical performance of the method is very promising and hopefully will ease your concerns, especially since our method still performs well under the sparse setting that is not to our advantage.
>
>
> ----
>
> > **Q**: Literature review.
>
> **A**: Thanks for pointing that out. We have included more relevant papers although it is hard to be exhaustive.

---

> ### Author Response · Authors · 2022-11-23
> **Clarification on the revision**
>
> Thank you for raising the score and for your appreciation of our efforts. We would like to point out that the revision is more about the presentation and interpretation of the theorem with additional simulation studies as requested.
>
> Specifically, for the theoretical section (Section 3.2), we moved the non-asymptotic result from the proof in the appendix (equation (14) in the original version or equation (15) in the revised version) to the main theorem since reviewers argued that the original asymptotic statement is not informative enough, especially in terms of the dependency between the dimensions and the rank. We also added remarks to address the interpretation concerns raised by the reviewers. We indeed improve the rates, which is only due to a more careful calculation. The proof ideas and techniques remains the same.
>
> We hope this addresses your concerns.

---

### Official Review · Reviewer_t2CV · 2022-10-25

**Confidence:** 3
**Correctness:** 4
**Technical Novelty And Significance:** 3
**Empirical Novelty And Significance:** 3
**Recommendation:** 6

**Clarity, Quality, Novelty And Reproducibility:**

The paper is largely clear, with all the results clearly stated. While empirical results are relatively lacking, the description is detailed enough for reproducibility.

**Strength And Weaknesses:**

Strength:

1. The paper is well-written, with all the sections coherently organized. The contents are largely easy to follow.

2. The intuition and the generalization value of the Hi-CCAB algorithm are also well-explained.

3. Empirical analysis of the algorithm is consistent with the theoretical claims.

Weaknesses:

1. While the overall picture/intuition of the algorithm, it is unclear how the $T^{-\frac{1}{13}}$ bound is achieved by the descriptions of the paper alone. Since the validity of the algorithm hinges on effective SVD/PCA algorithm, it would be helpful to specify(at least on a high-level) what is meant by 'carefully leveraging the independent parts of dependent variables' and what method is actually used.

2. The empirical section is a bit lacking. The only benchmark being used is the naive algorithm. While current results do suggest that the Hi-CCAB converges, it is unclear how it compares to other heuristics in bandit literature (e.g. $\epsilon$-greedy or UCB-based algorithms), and a comparison between well-known algorithms will be helpful too.


**Summary Of The Paper:**

This paper proposes a new model for contextual bandit problems with high dimensional contextual vector space and action space. Making use of singular value decomposition techniques over low-rank representation matrix, the Hi-CCAB algorithm in essence solves a regularized regression problem to find the best empirical parameter estimates till the end of horizon. Theoretical analysis provides guarantee on the Hi-CCAB algorithm, and empirical evaluation on real-time datasets corroborate with the claims.

**Summary Of The Review:**

My rating of this paper would be a 6, although I'd be happy to adjust my scores if the author address my concerns.

---

> ### Author Response · Authors · 2022-11-19
> **Response to Reviewer t2CV**
>
> Thank you for your positive review and for helping us to improve. We made the following two major changes in our revision to address your questions and comments.
>
> > **Q**: How the $T^{-1/13}$ bound is achieved? What is meant by "carefully leveraging the independent parts of dependent variables and what method is used?"
>
> **A**:
> We updated Section 3.2 for the theoretical results. We improve the bound to the order of $T^{-{2\over 15}}$. We also provide a proof sketch and a couple of remarks to elaborate on the techniques we used and the interpretation of the theorem.
>
> In particular,
> there are two major steps in the proof: (1) bounding the estimation error for the low-rank representation matrix estimator; (2) bounding the expected cumulative regret.
>
> (1) Bounding the estimation error of $\widehat{\boldsymbol{\Theta}}_t$ with a high probability bound. Denote $\delta\boldsymbol{\Theta}_t = \widehat{\boldsymbol{\Theta}}_t - \boldsymbol{\Theta}$. We show that for a large $t$,
>
> $$P\left(||\delta \boldsymbol{\Theta}_t||_F \le {3\over T^{2\over 15}} {\sigma\sqrt{d_x+1} \over \sqrt{L}h^2} + 6\lambda_0{\sqrt{2d}\over h^2T^{1\over 6}}\right) \le 1 - \left( {3\over t} + {2\over t^2} + {2\over Lt} + {2\over L^3t^3} + {1\over t^{2\over 15}}  \right).$$
>
> Note that the action taken is based on previous estimators and affects the accuracy of future estimators, leading to lots of dependencies. The classical matrix completion results can no longer apply. Through careful use of conditional expectations, martingales, and empirical process we separate out different sources of randomness from the exploration and the covariates (i.e., $\delta_1,\cdots, \delta_t, x_{1,\cdot},\cdots,x_{t,\cdot}$) to derive the bounds. Lemma 1 establishes a restricted-strong-convexity-type result of the sum of squares in the objective function. Lemma 2 establishes a Lipschitz-type result of the sum of squares in the objective function.  Further analysis of the nuclear-norm-penalized sum of squares with the two lemmas and low-rank properties gives the tail bound of the estimation error.
>
> (2)
> Bounding the time-averaged expected regret. Let $Q_t = \{||\delta \boldsymbol{\Theta}_t||_F \le {3\over T^{2\over 15}} {\sigma\sqrt{d_x+1} \over \sqrt{L}h^2} + 6\lambda_0{\sqrt{2d}\over h^2T^{1\over 6}}  \}$ be the event such that $\delta \boldsymbol{\Theta}_t$ is bounded. We know that from the first step, for large $t$, $P(Q_t^c) \le {3\over t} + {2\over t^2} + {2\over Lt} + {2\over L^3t^3} + {1\over t^{2\over 15}} $. Consider the expectation of the regret on $Q_t$ and $Q_t^c$ separately and both terms vanish with $t$ at the polynomial rate.
>
> ----
>
> > **Q**: How Hi-CCAB compares to other heuristics in bandit literature (e.g. $\epsilon$-greedy or UCB-based algorithms)?
>
> **A**: We agree that it would be helpful to compare with other well-known heuristics in bandit literature.
> Most bandit algorithms, however, are not applicable to our case study
> since the total possible combination of product assortment is very high-dimensional--there are 176 products in total and the maximum number of available slots is 30 leading to $176 \choose 30$ combinations. In addition, the arm space for the prices is continuous.
>
> To address your concerns, we evaluate Hi-CCAB under some classical bandit models that are actually special cases of ours as claimed in Proposition 1.
> In particular, we compare Hi-CCAB with  LinUCB (Liet al., 2010), Lasso Bandit (Bastani \& Bayati, 2020),  NeuralUCB (Zhou et al., 2020), and EE-Net (Ban et al., 2022) in Section 4 under the multi-armed bandit setup and we consider a non-sparse case and a sparse case.
> As shown in our simulation results, Hi-CCAB outperforms the other three algorithms in the non-sparse case
> and performs well in the sparse case, which is not to the advantage of Hi-CCAB.
> It only falls short of the Lasso bandit that is specifically designed for the sparse setting when the number of arms is relatively small. It still outperforms all the others when the number of arms becomes large.
>
>
> We hope our response and revision clarify your concerns and questions and that you will consider raising your score. Thank you!

---

### Author Response · Authors · 2022-11-19
**Paper revision**

We very much appreciate all the reviewers for their valuable feedback.
The reviews are tremendously helpful to improve the paper and
we spent much time and a major effort to address all the comments, questions, and concerns.
We believe the revision has brought the paper to a much better level and
has reinforced our claims in the paper.

All major modifications in the pdf file are highlighted in blue for ease of reading.
In particular, we have made the following majors changes as requested:

* In Section 3.2 theoretical results, we improved Theorem 1 and provide a proof sketch and interpretations for Theorem 1. In particular, we
    * improved convergence rate and provide a non-asymptotic result that explicitly shows the dependence in the theorem on all important factors including the rank of the representation matrix and the dimensions of the arm space and the covariate space;
    * provide more intuition on our algorithm by interpreting our theoretical result;
    * explain the proof ideas, including how we leverage the low-rank assumption to overcome the high dimensionalities of the arm space and the covariate space;
    * give interpretations to our result on the rate of time-averaged expected cumulative regret, including the dependency on the rank of the representation matrix as well as the dimensions of the arm space and contextual space.

* In Section 4, we provide additional simulation studies to compare Hi-CCAB with the classical LinUCB and the state-of-the-art methods including the Lasso bandit and neuralUCB.

* Clarifications on wordings requested by reviewers.

* Clarifications on our contributions.

We would like to re-emphasize the contribution of our model. The bandit problem with both a high-dimensional continuum arm space *and* a high-dimensional covariate space largely remains unsolved. It is an important question of both practical and theoretical interest.
The existing frameworks are suitable for bandit problems with either high-dimensional covariate action space or high-dimensional action space, but not both.
To our knowledge, our model is the first to formulate the high-dimensional continuum armed and high-dimensional contextual bandit problem.

We will address the questions and comments to each reviewer below.

---

### Decision · Program_Chairs · 2023-01-20

**Decision:**

Reject

**Justification For Why Not Higher Score:**

The reviewers share concerns on (1) the design of exploration and the regret bound are not clearly explained; (2) lacking experiments; and (3) literature review should be improved.  While the authors improved the paper and one reviewer increased score from 3 to 5, the overall rating is still lower than borderline.

**Justification For Why Not Lower Score:**

N/A

**Metareview: Summary, Strengths And Weaknesses:**

The paper considers the problem of contextual bandits with both a high-dimensional action space and a high-dimensional covariate space.  Assuming the existence of a low-rank representation matrix, the authors proposed Hi-CCAB algorithm that applies low-rank matrix estimator (SVD) and provided regret bound of the proposed algorithm. The reviewers agree that the considered problem is interesting. However, reviewers raised shared concerns including (1) the design of exploration and the regret bound ($O(T^{13/15})$ in the revised version) are not clearly explained; (2) lacking experiments; and (3) literature review should be improved. The authors have improved some of the aspects during response period but it requires further major revision. I therefore recommend rejecting the paper. Given the interesting problem setting, the authors are encouraged to revise the paper according to reviews and submit it to the next venue.